# The Gamma-Topp-Leone-Type II-Exponentiated Half Logistic-G Family of Distributions with Applications

**Broderick Oluyede \***[ID] **and Thatayaone Moakofi**[ID]

Department of Mathematics and Statistical Sciences, Botswana International University of Science and Technology, Palapye 10071, Botswana; thatayaone.moakofi@studentmail.biust.ac.bw
* Correspondence: boluyede@georgiasouthern.edu or oluyedeo@biust.ac.bw

**Abstract:** The new Ristić and Balakhrisnan or Gamma-Topp-Leone-Type II-Exponentiated Half Logistic-G (RB-TL-TII-EHL-G) family of distributions is introduced and investigated in this paper. This work derives and studies some of the main statistical characteristics of this new family of distributions. The maximum likelihood estimation technique is used to estimate the model parameters, and a simulation study is used to assess the consistency of the estimators. Applications to three real-life datasets from various fields show the value and adaptability of the new RB-TL-TII-EHL-G family of distributions. From our results, it is evident that the new proposed distribution is flexible enough to characterize datasets from different fields compared to several other existing distributions in the literature.

**Keywords:** gamma generator; Topp-Leone distribution; type II distribution; exponentiated-G distribution; half logistic distribution; likelihood function; goodness-of-fit statistics

## 1. Introduction

Probability distributions play an important role in statistical modeling and analysis in different fields including engineering, medicine, and life sciences. However, classical lifetime distributions such as the exponential distribution, Rayleigh distribution, Pareto distribution, and Weibull distribution have a limited range of behavior when it comes to modeling new varieties of real datasets. Hence, researchers have been increasingly interested in generating new families with high flexibility to act as alternatives to available distributions. Some of the newly generated distributions in the literature include the following: the extended exponential-Weibull distribution by [1], the beta Burr-type X distribution by [2], theheavy-tailed log-logistic distribution by [3], the half-Cauchy generalized exponential distribution by [4], the novel alpha power Gumbel distribution by [5], the Weibull-G family of probability distributions by [6], the type II quasi-Lambert-G family of probability distributions by [7], the novel-G family of distributions by [8], thePoisson reciprocal Rayleigh family of distributions by [9], a quasi-Poisson Topp-Leone generated-G family of distributions by [10], the Poisson generated family of distributions by [11], the Gompertz-G family of distributions by [12], the exponentiated Weibull-H family of distributions by [13], the power Lindley-G family of distributions by [14], the Marshall-Olkin odd power generalized Weibull-G family of distributions by [15], the exponential T-X family of distributions by [16], the type II exponentiated half logistic Topp-Leone-Marshall-Olkin-G family of distributions by [17], the type II exponentiated half logistic generated family of distributions by [18], type II exponentiated half logistic-Topp-Leone-G power series class of distributions by [17], the type II half logistic exponential distribution by [19], and the type II exponentiated half logistic-Gompertz Topp-Leone-G family of distributions by [20], to mention a few.

The gamma transformation has been used to extend various distributions available in the literature. The following are some generated distributions via the gamma transformation: the unit gamma-G class of distributions by [21], the Zografos-Balakrishnan-G

family of distributions by [22], the gamma odd power generalized Weibull-G family of distributions by [23], the gamma-generalized inverse Weibull distribution by [24], the Ristić and Balakrishnan Lindley-Poisson distribution by [25], the gamma odd Burr III-G family of distributions by [26], the Zografos-Balakrishnan Burr XII distribution by [27], the Zografos-Balakrishnan Lindley distribution by [28], the gamma odd Burr X-G family of distributions by [29], the gamma Lindley distribution by [30], the gamma Weibull-G family of distributions by [31], the gamma Kumaraswamy-G family of distributions by [32], the gamma extended Weibull family of distributions by [33], and the gamma-Rayleigh distribution by [34], to mention a few.

Our basic motivations lie in the flexibility of the new family of distributions to model both monotonic and non-monotonic hazard rate functions by capturing different shapes; the ability of the new model to provide better fits than the baseline and several extended distributions available in the literature; and the applicability of the special cases of the new family of distributions in real-life scenarios. Another interesting part is the role played by the extra shape parameter(s) by introducing skewness and modulating the weight of the tails of any baseline distribution.

Throughout this paper, we will set

$$K_G\left(x; a, \underline{\psi}\right) = \left(\frac{1 - G(x; \underline{\psi})}{1 + G(x; \underline{\psi})}\right)^{2a}.$$

The cumulative distribution function (cdf) and probability density function (pdf) of the new Topp-Leone-type II exponentiated half logistic-G family of distributions are given by

$$F(x; b, a, \underline{\psi}) = \left[1 - K_G\left(x; a, \underline{\psi}\right)\right]^b$$

and

$$f(x; b, a, \underline{\psi}) = 4ab\left[1 - K_G\left(x; a, \underline{\psi}\right)\right]^{b-1}\left(1 - G(x; \underline{\psi})\right)^{2a-1}$$
$$\times \frac{g(x; \underline{\psi})}{\left[1 + G(x; \underline{\psi})\right)\right]^{2(a+1)-1}},$$

for $b, a > 0$ and baseline parameter vector $\underline{\psi}$ (see [35] for additional details).

The gamma generator proposed by [36] has the cdf and pdf given by

$$F_{RB}(x; \delta) = 1 - \frac{1}{\Gamma(\delta)}\int_0^{-\log(G(x))} t^{\delta-1-t}dt, \quad \delta > 0$$

and

$$f_{RB}(x; \delta) = \frac{1}{\Gamma(\delta)}[-\log(G(x))]^{\delta-1}g(x), \quad x \in \mathbb{R},$$

where $G(x)$ is the baseline cdf and $\delta > 0$ is a shape parameter.

This article is structured as follows: Section 2 presents the proposed family and its sub-families. Section 3 contains the linear representation of the pdf of the RB-TL-TII-EHL-G family of distributions, the reliability function and the quantile function. We derive some of the statistical and mathematical properties under Section 4. Monte Carlo simulation results are given in Section 5. Some of the special cases are presented under Section 6. In Section 7, we present results on applications using real-life data to demonstrate the applicability and flexibility of the fitted model, and finally, we give concluding remarks under Section 8.

## 2. Gamma-Topp-Leone-Type II-Exponentiated Half Logistic-G (RB-TL-TII-EHL-G) Distribution

In this section, we introduce the new family of distributions with the cdf and pdf given by

$$
F_{RB-TL-TII-EHL-G}(x; \delta, a, b, \underline{\psi}) = 1 - \frac{1}{\Gamma(\delta)} \int_0^{-\log\left(\left[1-K_G(x;a,\underline{\psi})\right]^b\right)} t^{\delta-1-t} dt
$$

$$
= 1 - \frac{\gamma\left(\delta, -\log\left(\left[1 - K_G\left(x; a, \underline{\psi}\right)\right]^b\right)\right)}{\Gamma(\delta)}, \tag{1}
$$

and

$$
\begin{aligned}
f_{RB-TL-TII-EHL-G}(x; \delta, a, b, \underline{\psi}) &= \frac{4ab}{\Gamma(\delta)} \left[-\log\left(\left[1 - K_G\left(x; a, \underline{\psi}\right)\right]^b\right)\right]^{\delta-1} \\
&\times \left[1 - K_G\left(x; a, \underline{\psi}\right)\right]^{b-1} \left(1 - G(x; \underline{\psi})\right)^{2a-1} \\
&\times \frac{g(x; \underline{\psi})}{\left[1 + G(x; \underline{\psi})\right]^{2(a+1)-1}}, \tag{2}
\end{aligned}
$$

for $\delta, a, b > 0$ and parameter vector $\underline{\psi}$. The parameters $\delta, a$ and $b$ are shape parameters. The pdf helps us visualize shape and characteristics of the distribution, while the cdf provides a more inclusive view of probabilities and allows for various calculations and comparisons. We set $F_{RB-TL-TII-EHL-G}(x; \delta, a, b, \underline{\psi}) = F(x; \delta, a, b, \underline{\psi})$ and $f_{RB-TL-TII-EHL-G}(x; \delta, a, b, \underline{\psi}) = f(x; \delta, a, b, \underline{\psi})$, respectively.

*Sub-Families*

- When $\delta = 1$, we obtain the Topp-Leone-Type II-Exponentiated Half Logistic-G (TL-TII-EHL-G) family of distributions with the cdf

$$
F(x; b, a, \underline{\psi}) = \left[1 - K_G\left(x; a, \underline{\psi}\right)\right]^b,
$$

for $a, b > 0$, and parameter vector $\underline{\psi}$ (see [35]).

- When $a = 1$, we obtain the Gamma-Topp-Leone-Type II-Half Logistic-G (RB-TL-TII-HL-G) family of distributions with the cdf

$$
F(x; \delta, b, \underline{\psi}) = 1 - \frac{\gamma\left(\delta, -\log\left(\left[1 - \left(\frac{1-G(x;\underline{\psi})}{1+G(x;\underline{\psi})}\right)^2\right]^b\right)\right)}{\Gamma(\delta)},
$$

for $\delta, b > 0$, and parameter vector $\underline{\psi}$. This is a new family of distributions.

- When $b = 1$, we obtain the Gamma-Type II-Exponentiated Half Logistic-G (RB-TII-EHL-G) family of distributions with the cdf

$$
F(x; \delta, a, \underline{\psi}) = 1 - \frac{\gamma\left(\delta, -\log\left(\left[1 - K_G\left(x; a, \underline{\psi}\right)\right]\right)\right)}{\Gamma(\delta)},
$$

for $\delta, a > 0$, and parameter vector $\underline{\psi}$. This is a new family of distributions.

- When $a = b = 1$, we obtain the Gamma-Type II-Half Logistic-G (RB-TII-HL-G) family of distributions with the cdf

$$F(x; \delta, \underline{\psi}) = 1 - \frac{\gamma \left( \delta, -\log \left( \left[ 1 - \left( \frac{1 - G(x;\underline{\psi})}{1 + G(x;\underline{\psi})} \right)^2 \right] \right) \right)}{\Gamma(\delta)},$$

  for $\delta$ and parameter vector $\underline{\psi}$. This is a new family of distributions.

- When $\delta = a = 1$, we obtain the Topp-Leone-Type II-Half Logistic-G (TL-TII-HL-G) family of distributions with the cdf

$$F(x; b, \underline{\psi}) = \left[ 1 - \left( \frac{1 - G(x; \underline{\psi})}{1 + G(x; \underline{\psi})} \right)^2 \right]^b,$$

  for $b > 0$ and parameter vector $\underline{\psi}$. This is a new family of distributions.

- When $\delta = b = 1$, we obtain the Type II-Exponentiated Half Logistic-G (TII-EHL-G) family of distributions with the cdf

$$F(x; a, \underline{\psi}) = 1 - K_G \left( x; a, \underline{\psi} \right),$$

  for $a > 0$ and parameter vector $\underline{\psi}$ (see [18]).

- When $\delta = a = b = 1$, we obtain the new family of distributions with the cdf

$$F(x; \underline{\psi}) = 1 - \left( \frac{1 - G(x; \underline{\psi})}{1 + G(x; \underline{\psi})} \right)^2,$$

  for parameter vector $\underline{\psi}$.

## 3. Expansion of Density Function

In this section, we will derive the series expansion of the density function. Let $y = K_G \left( x; a, \underline{\psi} \right)$, and consider the series expansion $(-\log(1 - y)) = \sum_{i=0}^{\infty} \frac{y^{i+1}}{i + 1}$ and the following generalized binomial series expansion

$$(1 + z)^{-t} = \sum_{k=0}^{\infty} (-1)^k \binom{t + k - 1}{k} z^k \quad \text{for} \quad |z| < 1, \quad and \quad t > 0,$$

and using the results on power series raised to a positive integer, by setting $a_s = \frac{1}{s+2}$, that is $(\sum_{s=0}^{\infty} a_s y^s)^m = \sum_{s=0}^{\infty} b_{s,m} y^s$, we obtain

$$
\begin{aligned}
\left[ -\log \left( \left[ 1 - K_G \left( x; a, \underline{\psi} \right) \right]^b \right) \right]^{\delta - 1} &= b^{\delta-1} y^{\delta-1} \left[ \sum_{m=0}^{\infty} \binom{\delta - 1}{m} y^m \left( \sum_{s=0}^{\infty} \frac{y^s}{s + 2} \right)^m \right] \\
&= b^{\delta-1} y^{\delta-1} \left[ \sum_{m=0}^{\infty} \binom{\delta - 1}{m} y^m \sum_{s=0}^{\infty} b_{s,m} y^s \right] \\
&= b^{\delta-1} \left[ \sum_{m,s=0}^{\infty} \binom{\delta - 1}{m} b_{s,m} y^{m+s+\delta-1} \right],
\end{aligned}
$$

(see [37] for details), where $b_{s,m} = (s a_0)^{-1} \sum_{l=1}^{s} [m(l + 1) - s] a_l b_{s-l,m}$, $b_{0,m} = a_0^m$. The pdf of RB-TL-TII-EHL-G distribution can now be written as

$$f(x; \delta, a, b, \underline{\psi}) = \sum_{v=0}^{\infty} \omega_{v+1} g_{v+1}(x; \underline{\psi}), \tag{3}$$

(see Appendix A section for details of the derivation), where

$$
\begin{aligned}
\omega_{v+1} ={} & \frac{4ab}{\Gamma(\delta)} b^{\delta-1} \sum_{m,s,i,j,k,h=0}^{\infty} \binom{\delta-1}{m}\binom{b-1}{i} b_{s,m}(-1)^{i+j+p+v} \\
& \times \binom{2a[(m+s+\delta)+i]-1}{j}\binom{2a[(m+s+\delta)+i]+k}{k} \\
& \times \binom{j+k}{h}\binom{h}{v}\frac{1}{v+1},
\end{aligned}
$$

(4)

and $g_{v+1}(x;\underline{\psi}) = (v+1)G^v(x;\underline{\psi})g(x;\underline{\psi})$ is the exponentiated-G (Exp-G) pdf with power parameter $(v+1)$. Consequently, the mathematical and statistical properties of the RB-TL-TII-EHL-G family of distributions follow directly from those of the Exp-G family of distributions.

*Reliability, Hazard Rate and Quantile Functions*

The survival function gives us the overall survival probabilities over time, while the hazard rate function (hrf) provides information about the changing risk of the event occurrence. The survival function and hrf of the RB-TL-TII-EHL-G family of distributions are given, respectively, by

$$
S(x;\delta,a,b,\underline{\psi}) = \frac{\gamma\left(\delta, -\log\left(\left[1-K_G\left(x;a,\underline{\psi}\right)\right]^b\right)\right)}{\Gamma(\delta)},
$$

(5)

and

$$
\begin{aligned}
h(x;\delta,a,b,\underline{\psi}) ={} & 4ab\left[-\log\left(\left[1-K_G\left(x;a,\underline{\psi}\right)\right]^b\right)\right]^{\delta-1} \\
& \times \left[1-K_G\left(x;a,\underline{\psi}\right)\right]^{b-1}\left(1-G(x;\underline{\psi})\right)^{2a-1} \\
& \times \frac{g(x;\underline{\psi})}{\left[1+G(x;\underline{\psi}))\right]^{2(a+1)-1}} \\
& \times \left[\gamma\left(\delta, -\log\left(\left[1-K_G\left(x;a,\underline{\psi}\right)\right]^b\right)\right)\right]^{-1},
\end{aligned}
$$

(6)

for $\delta, a, b > 0$, and parameter vector $\underline{\psi}$. The quantile function is a very useful statistical tool when it comes to generating random numbers. It is also important when obtaining other statistical measures such as median, skewness and kurtosis. It is obtained by inverting the cdf, that is,

$$
1 - \frac{\gamma\left(\delta, -\log\left(\left[1-K_G\left(x;a,\underline{\psi}\right)\right]^b\right)\right)}{\Gamma(\delta)} = u,
$$

(7)

for $0 \le u \le 1$. Note that Equation (7) can be written as

$$
G(x;\underline{\psi}) = \frac{1-\left(1-\left[\exp(-\gamma^{-1}[\delta,\Gamma(\delta)(1-u)])\right]^{\frac{1}{b}}\right)^{\frac{1}{2a}}}{\left[1+\left(1-\left[\exp(-\gamma^{-1}[\delta,\Gamma(\delta)(1-u)])\right]^{\frac{1}{b}}\right)^{\frac{1}{2a}}\right]}.
$$

Therefore, the quantile function of the RB-TL-TII-EHL-G family of distributions is given by

$$Q_X(u) = G^{-1}\left[\frac{1 - \left(1 - [\exp(-\gamma^{-1}[\delta, \Gamma(\delta)(1-u)])]^{\frac{1}{b}}\right)^{\frac{1}{2a}}}{\left[1 + \left(1 - [\exp(-\gamma^{-1}[\delta, \Gamma(\delta)(1-u)])]^{\frac{1}{b}}\right)^{\frac{1}{2a}}\right]}\right]. \quad (8)$$

## 4. Mathematical Properties

In this section, we obtain several mathematical and statistical properties of the RB-TL-TII-EHL-G family of distributions. The properties presented include moments, moment-generating function, moments of residual and reversed residual life, Rényi Entropy, distribution of $r^{th}$-order statistics and stochastic ordering. Let the pdf $f(x; \delta, a, b, \underline{\psi})$ be written as $f(x)$ throughout this section.

### 4.1. Moments and Generating Function

Let $Y_{v+1} \sim Exponentiated - G(v+1, \underline{\psi})$; then, the $n^{th}$ raw moment $\mu'_n$ of the RB-TL-TII-EHL-G family of distributions is given by

$$\mu'_n = E(X^n) = \int_{-\infty}^{\infty} x^n f(x) dx = \sum_{v=0}^{\infty} \omega_{v+1} E(Y_{v+1}^n),$$

where $E(Y_{v+1}^n)$ is the $n^{th}$ moment of $Y_{v+1}$ and $\omega_{v+1}$ is given by Equation (4). The moment-generating function (MGF), for $|t| < 1$, is given by:

$$M_X(t) = \sum_{v=0}^{\infty} \omega_{v+1} M_{v+1}(t),$$

where $M_{v+1}(t)$ is the mgf of $Y_{v+1}$ and $\omega_{v+1}$ is given by Equation (4).

### 4.2. Moment of Residual and Reversed Residual Life

Moments of the residual life distribution are used to obtain the mean, variance and coefficient of variation of residual life which are extensively used in reliability analysis. The $s^{th}$ moment of the residual life, say $\phi_s(t)$ of a random variable $X$, is

$$\phi_s(t) = E\left[(X-t)^s \mid X > t\right] = \frac{1}{\overline{F}(t)} \int_t^{\infty} (x-t)^s f(x) dx.$$

Consequently, $\phi_s(t)$ for the RB-TL-TII-EHL-G distribution is given as follows:

$$\phi_s(t) = \frac{1}{\overline{F}(t)} \sum_{v,p=0}^{\infty} \binom{s}{p} (-t)^{s-p} \omega_{v+1} \int_t^{\infty} x^p g_{v+1}(x; \underline{\psi}) dx, \quad (9)$$

where $\omega_{v+1}$ is as defined in Equation (4) and $g_{v+1}(x; \underline{\psi})$ denotes the Exp-G distribution with power parameter $(v+1)$. The mean remaining life (life expectancy at age $t$) of the RB-TL-TII-EHL-G family of distributions follows from the above formula with $s = 1$.

The $s^{th}$ moment of the reversed residual life, say $U_s(t)$ of a random variable $X$, is

$$U_s(t) = E\left[(t-X)^s \mid X \leq t\right] = \frac{1}{F(t)} \int_0^t (t-x)^s f(x) dx.$$

Subsequently, $U_s(t)$ for the RB-TL-TII-EHL-G distribution is given as follows:

$$U_s(t) = \frac{1}{F(t)} \sum_{v,p=0}^{\infty} \binom{s}{p} (-t)^{s-p} \omega_{v+1} \int_0^t x^p g_{v+1}(x; \underline{\psi}) dx,$$

where $\omega_{v+1}$ is as defined in Equation (4) and $g_{v+1}(x; \underline{\psi})$ denotes the Exp-G distribution with power parameter $(v+1)$.

*4.3. Rényi Entropy*

In information theory, Rényi entropy is a measue of randomness or uncertainty in the system. The Rényi entropy of the RB-TL-EHL-G family of distributions is given by

$$I_R(q) = \frac{1}{1-q} \log\left(\int_0^{\infty} f^q(x) dx\right), q > 0 \text{ and } q \neq 1,$$

where

$$
\begin{aligned}
f^q(x) &= \frac{(4ab)^q}{(\Gamma(\delta))^q} \left[ -\log\left( \left[1 - K_G\left(x; a, \underline{\psi}\right)\right]^b \right) \right]^{q\delta - q} \\
&\times \left[1 - K_G\left(x; a, \underline{\psi}\right)\right]^{qb-q} \left(1 - G(x; \underline{\psi})\right)^{2aq-q} \\
&\times \frac{g^q(x; \underline{\psi})}{\left[1 + G(x; \underline{\psi})\right]^{2q(a+1)-q}}.
\end{aligned}
$$

Let $y = K_G\left(x; a, \underline{\psi}\right)$, and with the series expansion $(-\log(1 - y)) = \sum_{i=0}^{\infty} \frac{y^{i+1}}{i+1}$, as well as the results on power series raised to a positive integer given by $(\sum_{s=0}^{\infty} a_s y^s)^m = \sum_{s=0}^{\infty} b_{s,m} y^s$ (see [37] for details), we obtain

$$
\begin{aligned}
\left[ -\log\left( \left[1 - K_G\left(x; a, \underline{\psi}\right)\right]^b \right) \right]^{q\delta - q} &= b^{q\delta - q} y^{q\delta - q} \left[ \sum_{m=0}^{\infty} \binom{q\delta - q}{m} y^m \left( \sum_{s=0}^{\infty} \frac{y^s}{s+2} \right)^m \right] \\
&= b^{q\delta - q} y^{q\delta - q} \left[ \sum_{m=0}^{\infty} \binom{q\delta - q}{m} y^m \sum_{s=0}^{\infty} b_{s,m} y^s \right] \\
&= b^{q\delta - q} \left[ \sum_{m,s=0}^{\infty} \binom{q\delta - q}{m} b_{s,m} y^{m+s+q\delta - q} \right],
\end{aligned}
$$

where $b_{s,m} = (sa_0)^{-1} \sum_{l=1}^{s} [m(l+1) - s] a_l b_{s-l,m}$, $b_{0,m} = a_0^m$, so that $f^q(x)$ can be written as:

$$
\begin{aligned}
f^q(x) &= \frac{(4ab)^q}{(\Gamma(\delta))^q} b^{q\delta - q} \sum_{m,s,i,j,k,h,v=0}^{\infty} \binom{q\delta - q}{m} b_{s,m} (-1)^{i+j+h+v} \binom{qb - q}{i} \\
&\times \binom{2a[m+s+q\delta + i] - q}{j} \binom{2a[m+s+q\delta + i] + q + k - 1}{k} \\
&\times \binom{j+k}{h} \binom{h}{v} g^q(x; \underline{\psi}) \left( G(x; \underline{\psi}) \right)^v.
\end{aligned}
$$

(see Appendix A section for details of the derivation). Consequently, the Rényi entropy for the RB-TL-TII-EHL-G family of distributions is given by

$$
\begin{aligned}
I_R(q) &= \frac{1}{1-q} \log \left[ \frac{(4ab)^q}{(\Gamma(\delta))^q} b^{q\delta - q} \sum_{m,s,i,j,k,h,v=0}^{\infty} \binom{q\delta - q}{m} b_{s,m}(-1)^{i+j+h+v} \right. \\
&\quad \times \binom{qb - q}{i} \binom{2a[m+s+q\delta+i]-q}{j} \binom{2a[m+s+q\delta+i]+q+k-1}{k} \\
&\quad \times \left. \binom{j+k}{h} \binom{h}{v} \int_0^{\infty} \left( \left[1 + \frac{v}{q}\right] \left(G(x;\underline{\psi})\right)^{\frac{v}{q}} \left(g(x;\underline{\psi})\right) \right)^q dx \right] \\
&= \frac{1}{1-q} \log \left[ \sum_{v=0}^{\infty} \tau_v \exp((1-q)I_{REG}) \right],
\end{aligned}
\tag{10}
$$

for $q > 0$, $q \neq 1$, where $I_{REG} = \frac{1}{1-q} \log \left[ \int_0^{\infty} \left( \left[1 + \frac{v}{q}\right] \left(G(x;\underline{\psi})\right)^{\frac{v}{q}} \left(g(x;\underline{\psi})\right) \right)^q dx \right]$ is the Rényi entropy of Exp-G distribution with power parameter $(\frac{v}{q} + 1)$, and

$$
\begin{aligned}
\tau_v &= \frac{(4ab)^q}{(\Gamma(\delta))^q} b^{q\delta - q} \sum_{m,s,i,j,k,h=0}^{\infty} \binom{q\delta - q}{m} b_{s,m}(-1)^{i+j+h+v} \frac{1}{\left[1 + \frac{v}{q}\right]^q} \binom{qb - q}{i} \\
&\quad \times \binom{2a[m+s+q\delta+i]-q}{j} \binom{2a[m+s+q\delta+i]+q+k-1}{k} \binom{j+k}{h} \binom{h}{v}.
\end{aligned}
$$

Therefore, the Rényi entropy of the RB-TL-TII-EHL-G family of distributions can be obtained from those of the Exp-G family of distributions.

### 4.4. Order Statistics

Order statistics are a very useful statistical concept in probability and statistics. We see its applications in several fields including modeling insurance policies, auctions, optimizing production processes, car races, estimating parameters of distributions, and many more. Suppose $X_1, X_2, \ldots, X_n$ are independent and identically distributed random variables from the RB-TL-TII-EHL-G family of distributions. Then, the expression for the pdf of the $r^{th}$-order statistic from the RB-TL-TII-EHL-G distribution can be written as

$$
f_{r:n}(x) = \frac{n! f(x)}{(r-1)!(n-r)!} \sum_{p=0}^{n-r} (-1)^p \binom{n-r}{p} [F(x)]^{p+r-1}.
\tag{11}
$$

Note that

$$
\begin{aligned}
f(x)[F(x)]^{p+r-1} &= \sum_{z=0}^{\infty} \binom{p+r-1}{z} (-1)^z \frac{4ab}{(\Gamma(\delta))^{z+1}} \left[ -\log\left( \left[1 - K_G\left(x;a,\underline{\psi}\right)\right]^b \right) \right]^{\delta-1} \\
&\quad \times \left[1 - K_G\left(x;a,\underline{\psi}\right)\right]^{b-1} \left(1 - G(x;\underline{\psi})\right)^{2a-1} \\
&\quad \times \frac{g(x;\underline{\psi})}{\left[1 + G(x;\underline{\psi}))\right]^{2(a+1)-1}} \left[ \gamma\left(\delta, -\log\left( \left[1 - K_G\left(x;a,\underline{\psi}\right)\right]^b \right)\right) \right]^z,
\end{aligned}
$$

and using the results on the expansion of the density and the following power series for the incomplete gamma function (see [37]),

$$
\gamma(y, \delta) = \sum_{q=0}^{\infty} \frac{(-1)^q y^{q+\delta}}{(q+\delta)q!},
$$

we obtain

$$
\begin{aligned}
f(x)[F(x)]^{p+r-1} \;=\;& \sum_{z=0}^{\infty} \binom{p+r-1}{z} \left[ \sum_{q=0}^{\infty} \frac{(-1)^q}{(q+\delta)q!} \right]^z (-1)^z \frac{4ab}{(\Gamma(\delta))^{z+1}} \\
&\times \left[ -\log\left( \left[ 1 - K_G\left(x; a, \underline{\psi}\right) \right]^b \right) \right]^{z(q+\delta)+\delta-1} \\
&\times \left[ 1 - K_G\left(x; a, \underline{\psi}\right) \right]^{b-1} \left( 1 - G(x; \underline{\psi}) \right)^{2a-1} \frac{g(x; \underline{\psi})}{\left[ 1 + G(x; \underline{\psi}) \right)]^{2(a+1)-1}}.
\end{aligned}
$$

Now, following the same steps leading to Equation (3), we obtain

$$
f(x)[F(x)]^{p+r-1} = \sum_{v=0}^{\infty} a_{v+1} g_{v+1}(x; \underline{\psi}), \tag{12}
$$

where $g_{v+1}(x; \underline{\psi}) = (v+1)[G(x; \underline{\psi})]^v g(x; \underline{\psi})$ is the Exp-G pdf with the power parameter $(v+1)$ and parameter vector $\underline{\psi}$, and

$$
\begin{aligned}
a_{v+1} \;=\;& \sum_{z,m,s,i,j,k,h=0}^{\infty} \binom{p+r-1}{z} \left[ \sum_{q=0}^{\infty} \frac{(-1)^q}{(q+\delta)q!} \right]^z \frac{4ab(-1)^{z+i+j+p+v}}{(\Gamma(\delta))^{z+1}} b^{z(q+\delta)+\delta-1} \\
&\times \binom{z(q+\delta)+\delta-1}{m} \binom{b-1}{i} b_{s,m} \binom{j+k}{h} \binom{h}{v} \frac{1}{v+1} \\
&\times \binom{2a[(m+s+\delta)+i]-1}{j} \binom{2a[(m+s+\delta)+i]+k}{k}.
\end{aligned}
$$

Thus, by substituting (12) into (11), the pdf of the $r^{th}$-order statistic for the RB-TL-TII-EHL-G family of distributions can be written as

$$
f_{r:n}(x) \;=\; \frac{n!}{(r-1)!(n-r)!} \sum_{p=0}^{n-r} (-1)^p \binom{n-r}{p} \sum_{v=0}^{\infty} a_{v+1} g_{v+1}(x; \underline{\psi}). \tag{13}
$$

*4.5. Probability Weighted Moments (PWMs)*

The PWMs of a random variable $X$ are defined by

$$
\phi_{w,p} = E\left( X^w (F(X))^p \right) = \int_{-\infty}^{\infty} x^w F(x)^p f(x)\, dx.
$$

Using the results from the derivation of the distribution of the order statistics above, we note that

$$
\begin{aligned}
f(x)[F(x)]^p \;=\;& \sum_{z=0}^{\infty} \binom{p}{z} \left[ \sum_{q=0}^{\infty} \frac{(-1)^q}{(q+\delta)q!} \right]^z (-1)^z \frac{4ab}{(\Gamma(\delta))^{z+1}} \\
&\times \left[ -\log\left( \left[ 1 - K_G\left(x; a, \underline{\psi}\right) \right]^b \right) \right]^{z(q+\delta)+\delta-1} \\
&\times \left[ 1 - K_G\left(x; a, \underline{\psi}\right) \right]^{b-1} \left( 1 - G(x; \underline{\psi}) \right)^{2a-1} \frac{g(x; \underline{\psi})}{\left[ 1 + G(x; \underline{\psi}) \right)]^{2(a+1)-1}}.
\end{aligned}
$$

Now, following the same steps leading to Equation (3), we obtain

$$
f(x)[F(x)]^p = \sum_{v=0}^{\infty} C_{v+1} g_{v+1}(x; \underline{\psi}), \tag{14}
$$

where $g_{v+1}(x; \underline{\psi}) = (v+1)[G(x; \underline{\psi})]^v g(x; \underline{\psi})$ is the Exp-G pdf with the power parameter $(v+1)$ and parameter vector $\underline{\psi}$, and

$$
\begin{aligned}
C_{v+1} &= \sum_{z,m,s,i,j,k,h=0}^{\infty} \binom{p}{z} \left[ \sum_{q=0}^{\infty} \frac{(-1)^q}{(q+\delta)q!} \right]^z \frac{4ab(-1)^{z+i+j+p+v}}{(\Gamma(\delta))^{z+1}} b^{z(q+\delta)+\delta-1} \\
&\quad \times \binom{z(q+\delta)+\delta-1}{m} \binom{b-1}{i} b_{s,m} \binom{j+k}{h} \binom{h}{v} \frac{1}{v+1} \\
&\quad \times \binom{2a[(m+s+\delta)+i]-1}{j} \binom{2a[(m+s+\delta)+i]+k}{k}.
\end{aligned}
$$

Thus, the probability weighted moment of the RB-TL-TII-EHL-G family of distributions is given by

$$
\phi_{w,p} = E\big(X^w (F(X))^p\big) = \sum_{v=0}^{\infty} C_{v+1} \int_{-\infty}^{\infty} x^w g_{v+1}(x; \underline{\psi}). \tag{15}
$$

*4.6. Stochastic Orderings*

In probability and statistics, the notion of stochastic ordering for random variables is a useful concept. It quantifies the concept of one random variable being bigger than another [38]. The usual stochastic order, the hazard rate order, and likelihood ratio order are perhaps the best-known orders of distribution functions. In this subsection, we present likelihood ratio ordering.

Suppose $X$ and $Y$ are two random variables with the cdfs $F_X(t)$ and $F_Y(t)$, respectively, and $\overline{F}_X(t) = 1 - F_X(t)$ is the survival function. The random variable $X$ is said to be stochastically smaller than the random variable $Y$ if $\overline{F}_X(t) \leq \overline{F}_Y(t)$ or $F_X(t) \geq F_Y(t)$, $\forall t$. This is denoted by $X <_{st} Y$. The hazard rate order and likelihood ratio order are given by $X <_{hr} Y$ if $h_X(t) \geq h_Y(t)\ \forall t$, and $X <_{lr} Y$ if $\frac{f_X(t)}{f_Y(t)}$ is decreasing in t, $\forall t$. It is well established that $X <_{lr} Y \implies X <_{hr} Y \implies X <_{st} Y$ [38].

Now consider two independent random variables $X_1$ and $X_2$ following the RB-TL-TII-EHL-G family of distributions with $X_1 \sim f_1(x; \delta_1, a, b, \underline{\psi})$ and $X_2 \sim f_2(x; \delta_2, a, b, \underline{\psi})$, and their pdfs are given by

$$
\begin{aligned}
f_1(x) &= \frac{4ab}{\Gamma(\delta_1)} \left[ -\log\left( \left[ 1 - K_G\left(x; a, \underline{\psi}\right)\right]^b \right) \right]^{\delta_1-1} \left[ 1 - K_G\left(x; a, \underline{\psi}\right)\right]^{b-1} \\
&\quad \times \left(1 - G(x; \underline{\psi})\right)^{2a-1} \frac{g(x; \underline{\psi})}{\left[1 + G(x; \underline{\psi})\right)\right]^{2(a+1)-1}}
\end{aligned} \tag{16}
$$

and

$$
\begin{aligned}
f_2(x) &= \frac{4ab}{\Gamma(\delta_2)} \left[ -\log\left( \left[ 1 - K_G\left(x; a, \underline{\psi}\right)\right]^b \right) \right]^{\delta_2-1} \left[ 1 - K_G\left(x; a, \underline{\psi}\right)\right]^{b-1} \\
&\quad \times \left(1 - G(x; \underline{\psi})\right)^{2a-1} \frac{g(x; \underline{\psi})}{\left[1 + G(x; \underline{\psi})\right)\right]^{2(a+1)-1}},
\end{aligned} \tag{17}
$$

respectively. Then

$$
\frac{f_1(x)}{f_2(x)} = \frac{\Gamma(\delta_2)}{\Gamma(\delta_1)} \left[ -\log\left( \left[ 1 - K_G\left(x; a, \underline{\psi}\right)\right]^b \right) \right]^{\delta_1-\delta_2}. \tag{18}
$$

Differentiating Equation (18) with respect to $x$ yields

$$\frac{d}{dx}\left(\frac{f_1(x)}{f_2(x)}\right) = \frac{4ab(\delta_2 - \delta_1)\Gamma(\delta_2)}{\Gamma(\delta_1)}\left[-\log\left(\left[1 - K_G\left(x; a, \underline{\psi}\right)\right]^b\right)\right]^{\delta_1 - \delta_2 - 1}$$
$$\times \left[1 - K_G\left(x; a, \underline{\psi}\right)\right]^{b-1}\left(1 - G(x; \underline{\psi})\right)^{2a-1}$$
$$\times \frac{g(x; \underline{\psi})}{\left[1 + G(x; \underline{\psi})\right]^{2(a+1)-1}}\left(\left[1 - K_G\left(x; a, \underline{\psi}\right)\right]^b\right)^{-1}. \tag{19}$$

Consequently, $\frac{d}{dx}\left[\frac{f_1(x)}{f_2(x)}\right] < 0$ if $\delta_2 < \delta_1$. Thus, the likelihood ratio $X <_{lr} Y$ exists. Subsequently, since $X <_{lr} Y \implies X <_{hr} Y \implies X <_{st} Y$, the hazard rate order and stochastic order also hold.

*4.7. Maximum Likelihood Estimation*

There are several methods available in the literature for estimating unknown parameters of a probability distribution. Among them, the maximum likelihood method is the most commonly used. The maximum likelihood estimation (MLE) is a technique used in statistics to estimate the unknown parameters of an assumed probability distribution based on some experimental data. In this subsection, we use the maximum likelihood estimation method to obtain estimates of the parameters of the RB-TL-TII-EHL-G family of distributions. Suppose $x_1, x_2, \ldots\ldots, x_n$ is the random sample observed from the RB-TL-TII-EHL-G family of distributions with the vector of model parameters $\mathbf{\Delta} = (\delta, a, b, \underline{\psi})^T$. Then, the log-likelihood function $\ell_n = \ell_n(\mathbf{\Delta})$ for the parameters from the observed values has the form

$$\begin{aligned}\ell_n(\mathbf{\Delta}) &= n\ln(4ab) - n\ln(\Gamma(\delta)) + (\delta - 1)\sum_{i=1}^{n}\ln\left[-\log\left(\left[1 - \left(\frac{1 - G(x_i; \underline{\psi})}{1 + G(x_i; \underline{\psi})}\right)^{2a}\right]^b\right)\right]\\ &+ (b-1)\sum_{i=1}^{n}\left[1 - \left(\frac{1 - G(x_i; \underline{\psi})}{1 + G(x_i; \underline{\psi})}\right)^{2a}\right] + (2a-1)\sum_{i=1}^{n}\ln\left(1 - G(x_i; \underline{\psi})\right)\\ &- (2(a+1)-1)\sum_{i=1}^{n}\ln\left[1 + G(x_i; \underline{\psi})\right] + \sum_{i=1}^{n}\ln(g(x_i; \underline{\psi})). \end{aligned} \tag{20}$$

The elements of the score vector $U = \left(\frac{\partial\ell}{\partial\delta}, \frac{\partial\ell}{\partial a}, \frac{\partial\ell}{\partial b}, \frac{\partial\ell}{\partial\underline{\psi}_k}\right)$ are given in Appendix A.

The maximum likelihood estimates of the parameters, denoted by $\hat{\mathbf{\Delta}}$, are obtained by solving the nonlinear equation $\left(\frac{\partial\ell}{\partial\delta}, \frac{\partial\ell}{\partial a}, \frac{\partial\ell}{\partial b}, \frac{\partial\ell}{\partial\underline{\psi}_k}\right)^T = \mathbf{0}$. However, these equations are not in closed form; hence, they are solved by a numerical method such as the Newton–Raphson procedure.

## 5. Monte Carlo Simulation Results

In this section, a simulation study is used to assess the estimators of the parameters of the Gamma-Topp-Leone-type II-Exponentiated Half Logistic-Weibull (RB-TL-TII-EHL-W) distribution. Here, $N = 3000$ samples of size n = 25, 50, 100, 200, 400, 800, 1600 are generated from the RB-TL-TII-EHL-W distribution for different parameter values. For example, in Table 1, the true parameter values are chosen arbitrary to be (0.4, 2.0, 1.0, 0.6)

and (1.0, 1.0, 2.0, 0.6). The average bias (ABIAS) and root mean square errors (RMSE) for the estimated parameter, say, $\hat{\theta}$, for the MLE are calculated as follows:

$$ABias(\hat{\theta}) = \frac{\sum_{i=1}^{N} \hat{\theta}_i}{N} - \theta, \quad \text{and} \quad RMSE(\hat{\theta}) = \sqrt{\frac{\sum_{i=1}^{N} (\hat{\theta}_i - \theta)^2}{N}},$$

respectively. The simulation results are presented in Tables 1 and 2.

It can be observed in Tables 1 and 2 that the mean MLEs are close to the true values of the parameters, and the RMSEs decay toward zero as the sample size increases. This indicates that the maximum likelihood method works well for obtaining estimate of the model parameters of the RB-TL-TII-EHL-W distribution.

**Table 1.** Monte Carlo Simulation Results.

| Parameter | Sample Size | (0.4, 2.0, 1.0, 0.6) | | | (1.0, 1.0, 2.0, 0.6) | | |
|---|---|---|---|---|---|---|---|
| | | Mean | RMSE | ABIAS | Mean | RMSE | ABIAS |
| $\delta$ | 25 | 0.6877 | 3.1506 | 0.2877 | 1.3229 | 0.8782 | 0.3229 |
| | 50 | 0.3709 | 0.1177 | 0.0290 | 1.2402 | 0.6605 | 0.2402 |
| | 100 | 0.3787 | 0.0802 | 0.0212 | 1.1232 | 0.3570 | 0.1232 |
| | 200 | 0.3822 | 0.0603 | 0.0177 | 1.1006 | 0.2738 | 0.1006 |
| | 400 | 0.3922 | 0.0433 | 0.0077 | 1.0952 | 0.2100 | 0.0952 |
| | 800 | 0.3951 | 0.0365 | 0.0048 | 1.0925 | 0.2000 | 0.0925 |
| | 1600 | 0.3981 | 0.0274 | 0.0018 | 1.0491 | 0.1333 | 0.0491 |
| $a$ | 25 | 2.5400 | 1.3914 | 0.5400 | 1.1886 | 0.6130 | 0.1886 |
| | 50 | 2.3977 | 0.9355 | 0.3977 | 1.1187 | 0.4540 | 0.1187 |
| | 100 | 2.2693 | 0.7333 | 0.2693 | 1.0664 | 0.2516 | 0.0664 |
| | 200 | 2.2078 | 0.5503 | 0.2078 | 1.0298 | 0.1603 | 0.0298 |
| | 400 | 2.1536 | 0.4217 | 0.1536 | 1.0125 | 0.0996 | 0.0125 |
| | 800 | 2.1325 | 0.3502 | 0.1325 | 1.0010 | 0.0549 | 0.0010 |
| | 1600 | 2.0891 | 0.2558 | 0.0891 | 0.9990 | 0.0295 | -0.0009 |
| $b$ | 25 | 1.3057 | 0.8845 | 0.3057 | 2.7881 | 2.8433 | 0.7881 |
| | 50 | 1.2950 | 0.6806 | 0.2950 | 2.6862 | 1.5921 | 0.6862 |
| | 100 | 1.2521 | 0.4924 | 0.2521 | 2.6663 | 1.5291 | 0.6663 |
| | 200 | 1.2051 | 0.3896 | 0.2051 | 2.5486 | 1.2348 | 0.5486 |
| | 400 | 1.1270 | 0.2615 | 0.1270 | 2.3736 | 0.9899 | 0.3736 |
| | 800 | 1.0838 | 0.1916 | 0.0838 | 2.3047 | 0.8254 | 0.3047 |
| | 1600 | 1.0497 | 0.1160 | 0.0497 | 2.1826 | 0.5966 | 0.1826 |
| $\lambda$ | 25 | 1.6935 | 2.4098 | 1.0935 | 1.1408 | 1.4770 | 0.5408 |
| | 50 | 1.5659 | 2.1364 | 0.9659 | 1.0127 | 1.2357 | 0.4127 |
| | 100 | 1.4300 | 1.8248 | 0.8300 | 0.9742 | 1.1543 | 0.3742 |
| | 200 | 0.9253 | 1.0925 | 0.3253 | 0.7660 | 0.9116 | 0.1660 |
| | 400 | 0.8234 | 0.8786 | 0.2234 | 0.7619 | 0.8445 | 0.1619 |
| | 800 | 0.6774 | 0.5516 | 0.0774 | 0.6842 | 0.6534 | 0.0842 |
| | 1600 | 0.5944 | 0.3520 | 0.0055 | 0.6821 | 0.5891 | 0.0821 |

**Table 2.** Monte Carlo Simulation Results.

| Parameter | Sample Size | (1.3, 2.5, 1.3, 0.2) | | | (0.4, 2.0, 0.4, 0.6) | | |
|---|---|---|---|---|---|---|---|
| | | Mean | RMSE | ABIAS | Mean | RMSE | ABIAS |
| $\delta$ | 25 | 1.8712 | 1.0697 | 0.5712 | 0.5461 | 0.5260 | 0.1461 |
| | 50 | 1.5872 | 0.8045 | 0.2872 | 0.4494 | 0.2333 | 0.0494 |
| | 100 | 1.5855 | 0.7239 | 0.2855 | 0.4226 | 0.1363 | 0.0226 |
| | 200 | 1.4840 | 0.5620 | 0.1840 | 0.4178 | 0.0899 | 0.0178 |
| | 400 | 1.4472 | 0.4748 | 0.1472 | 0.4112 | 0.0667 | 0.0112 |
| | 800 | 1.3721 | 0.3075 | 0.0721 | 0.4102 | 0.0397 | 0.0102 |
| | 1600 | 1.3202 | 0.1560 | 0.0202 | 0.4070 | 0.0292 | 0.0070 |
| $a$ | 25 | 1.8861 | 1.2069 | 0.6138 | 2.5733 | 1.1101 | 0.5733 |
| | 50 | 1.9280 | 1.2898 | 0.5719 | 2.4370 | 0.8351 | 0.4370 |
| | 100 | 2.0105 | 1.2688 | 0.4894 | 2.3850 | 0.6897 | 0.3850 |
| | 200 | 2.1445 | 1.2661 | 0.3554 | 2.2559 | 0.5008 | 0.2559 |
| | 400 | 2.2502 | 1.1785 | 0.2497 | 2.1381 | 0.2995 | 0.1381 |
| | 800 | 2.4325 | 1.0837 | 0.0674 | 2.0626 | 0.1716 | 0.0626 |
| | 1600 | 2.5389 | 0.7505 | 0.0389 | 2.0387 | 0.1308 | 0.0387 |
| $b$ | 25 | 1.6020 | 2.0325 | 0.3020 | 1.1007 | 1.2914 | 0.7007 |
| | 50 | 1.5671 | 1.0407 | 0.2671 | 0.8581 | 0.8695 | 0.4581 |
| | 100 | 1.5539 | 0.7707 | 0.2539 | 0.7758 | 0.6698 | 0.3758 |
| | 200 | 1.5071 | 0.5967 | 0.2071 | 0.6940 | 0.5466 | 0.2940 |
| | 400 | 1.4872 | 0.5199 | 0.1872 | 0.5694 | 0.3332 | 0.1694 |
| | 800 | 1.4103 | 0.3495 | 0.1103 | 0.4961 | 0.2103 | 0.0961 |
| | 1600 | 1.3553 | 0.2097 | 0.0553 | 0.4691 | 0.1741 | 0.0691 |
| $\lambda$ | 25 | 1.9015 | 1.8505 | 1.7015 | 3.1446 | 3.9098 | 2.5446 |
| | 50 | 0.5098 | 1.6226 | 0.3098 | 2.0028 | 2.9795 | 1.4028 |
| | 100 | 0.2969 | 0.3970 | 0.0969 | 1.2378 | 1.6666 | 0.6378 |
| | 200 | 0.2472 | 0.1398 | 0.0472 | 0.8019 | 0.6485 | 0.2019 |
| | 400 | 0.2255 | 0.0846 | 0.0255 | 0.6875 | 0.2265 | 0.0875 |
| | 800 | 0.2169 | 0.0568 | 0.0169 | 0.6418 | 0.1240 | 0.0418 |
| | 1600 | 0.2132 | 0.0374 | 0.0132 | 0.6250 | 0.0722 | 0.0250 |

## 6. Special Cases

In this section, we specify the baseline distribution and define some special cases of the RB-TL-TII-EHL-G family of distributions.

*6.1. Gamma-Topp-Leone-Type II-Exponentiated Half Logistic-Weibull (RB-TL-TII-EHL-W) Distribution*

Consider the Weibull distribution with cdf and pdf given by $G(x; \lambda) = 1 - \exp(-x^\lambda)$ and $g(x; \lambda) = \lambda x^{\lambda-1} \exp(-x^\lambda)$, respectively, for $\lambda > 0$ and $x > 0$, as the baseline distribution; then, we obtain the RB-TL-TII-EHL-W distribution with cdf and pdf given by

$$F_{RB-TL-TII-EHL-W}(x; \delta, a, b, \lambda) = 1 - \frac{\gamma\left(\delta, -\log\left(\left[1 - \left(\frac{\exp(-x^\lambda)}{1 + (1 - \exp(-x^\lambda))}\right)^{2a}\right]^b\right)\right)}{\Gamma(\delta)},$$

and

$$
\begin{aligned}
f_{RB-TL-TII-EHL-W}(x; \delta, a, b, \lambda) &= \frac{4ab}{\Gamma(\delta)}\left[-\log\left(\left[1 - \left(\frac{\exp(-x^\lambda)}{1 + (1 - \exp(-x^\lambda))}\right)^{2a}\right]^b\right)\right]^{\delta-1} \\
&\times \left[1 - \left(\frac{\exp(-x^\lambda)}{1 + (1 - \exp(-x^\lambda))}\right)^{2a}\right]^{b-1} \left(\exp(-x^\lambda)\right)^{2a-1} \\
&\times \frac{\lambda x^{\lambda-1} \exp(-x^\lambda)}{[1 + (1 - \exp(-x^\lambda))]^{2(a+1)-1}},
\end{aligned}
$$

for $\delta, a, b, \lambda > 0$.

Figure 1 shows the plots of the pdf and the hrf of the RB-TL-TII-EHL-W distribution for different parameter values. The pdf can take several shapes including right-skewed, left-skewed, unimodal, J and reverse-J shapes. The RB-TL-TII-EHL-W hazard displays increasing, decreasing, bathtub, and upside-down bathtub shapes.

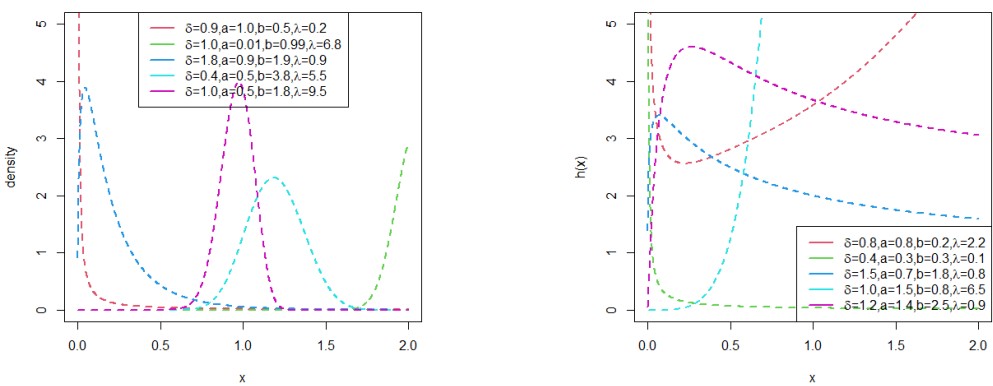

**Figure 1.** Plots of the pdf and hrf of the RB-TL-TII-EHL-W distribution.

Figure 2 shows the plots of skewness and kurtosis for the TL-TII-EHL-W distribution. We can see that the skewness become right-skewed and kurtosis is leptokurtic with increasing values of *a* and *b* and also with increasing values of *a* and $\lambda$.

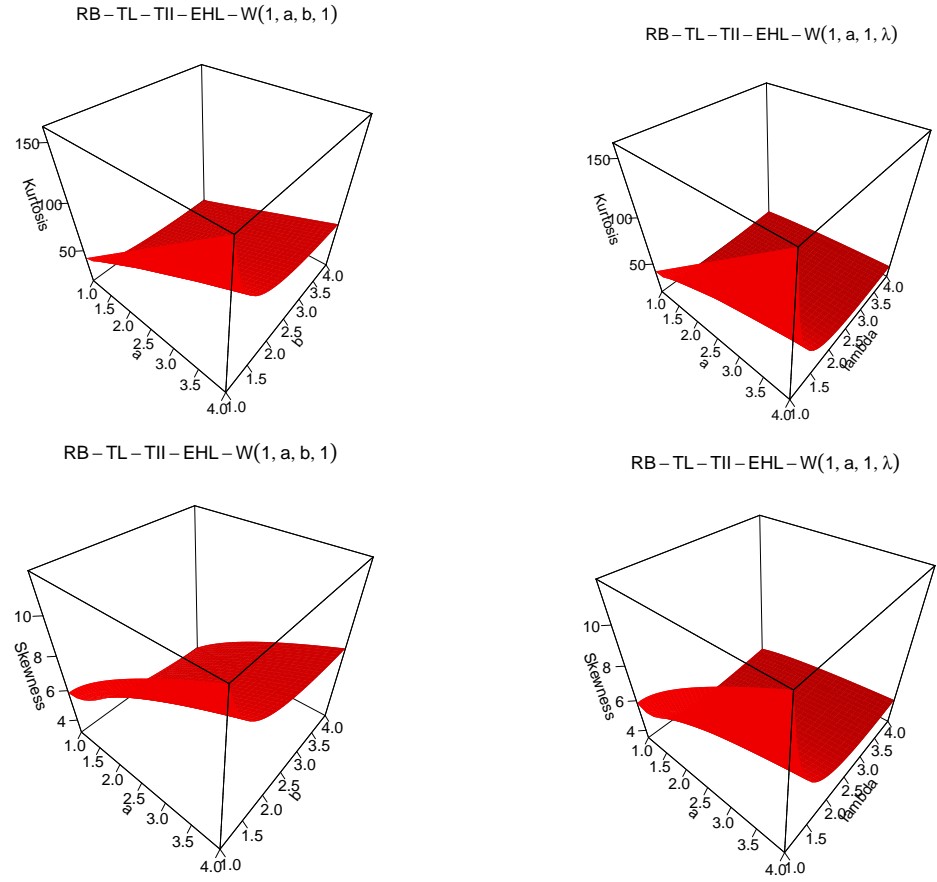

**Figure 2.** Three-dimensional (3D) plots of skewness and kurtosis for RB-TL-TII-EHL-W distribution.

*6.2. Gamma-Topp-Leone-Type II-Exponentiated Half Logistic-Log Logistic (RB-TL-TII-EHL-LLoG) Distribution*

Suppose the cdf and pdf of the baseline distribution are given by $G(x;c) = 1 - (1 + x^c)^{-1}$ and $g(x;\lambda,c) = cx^{c-1}(1+x^c)^{-2}$ for $c > 0$ and $x > 0$; then, the new RB-TL-TII-EHL-LLoG distribution has cdf and pdf given by

$$F_{RB-TL-TII-EHL-LLoG}(x;\delta,a,b,c) = 1 - \frac{\gamma\left(\delta, -\log\left(\left[1 - \left(\frac{(1+x^c)^{-1}}{1 + (1 - (1+x^c)^{-1})}\right)^{2a}\right]^b\right)\right)}{\Gamma(\delta)},$$

and

$$
\begin{aligned}
f_{RB-TL-TII-EHL-LLoG}(x;\delta,a,b,c) &= \frac{4ab}{\Gamma(\delta)}\left[-\log\left(\left[1 - \left(\frac{(1+x^c)^{-1}}{1 + (1 - (1+x^c)^{-1})}\right)^{2a}\right]^b\right)\right]^{\delta-1} \\
&\times \left[1 - \left(\frac{(1+x^c)^{-1}}{1 + (1 - (1+x^c)^{-1})}\right)^{2a}\right]^{b-1}\left((1+x^c)^{-1}\right)^{2a-1} \\
&\times \frac{cx^{c-1}(1+x^c)^{-2}}{[1 + (1 - (1+x^c)^{-1})]^{2(a+1)-1}},
\end{aligned}
$$

respectively, for $\delta, a, b, c > 0$.

Figure 3 shows the plots of pdf and hrf of RB-TL-TII-EHL-LLoG distribution, respectively. The pdf can take several shapes including right-skewed, left-skewed, unimodal and reverse-J shapes. The RB-TL-TII-EHL-LLoG hazard displays increasing, decreasing, upside-down bathtub, and bathtub followed by upside-down bathtub shapes.

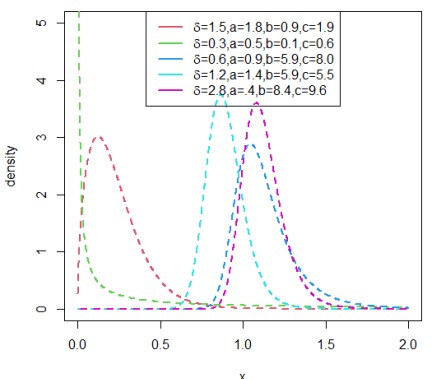
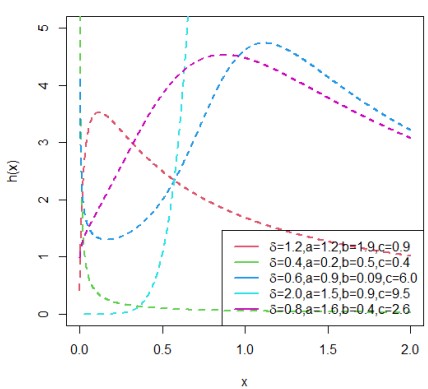

**Figure 3.** Plots of the pdf and hrf of the RB-TL-TII-EHL-LLoG distribution.

Figure 4 shows the plots of skewness and kurtosis for the RB-TL-TII-EHL-LLoG distribution. We can see that the skewness become right-skewed and kurtosis is leptokurtic with increasing values of $a$ and $b$ and also with increasing values of $a$ and $c$.

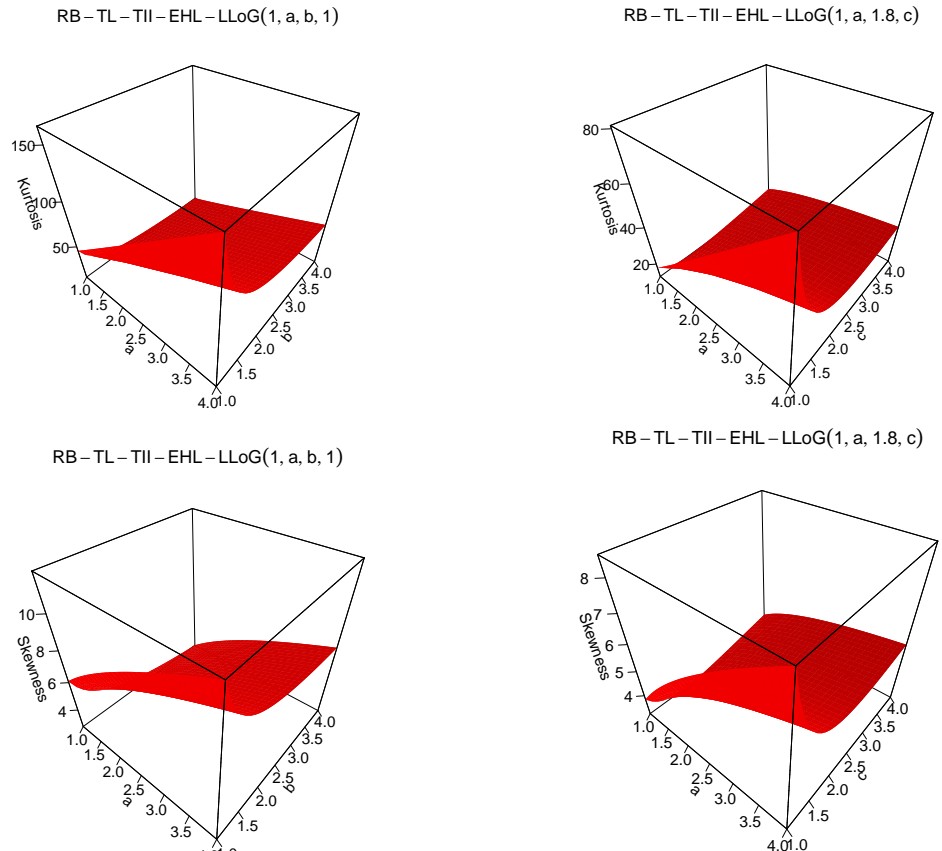

**Figure 4.** Three-dimensional (3D) plots of skewness and kurtosis for RB-TL-TII-EHL-LLoG distribution.

### 6.3. Gamma-Topp-Leone-Type II-Exponentiated Half Logistic-Kumaraswamy (RB-TL-TII-EHL-Kum) Distribution

If we consider the Kumaraswamy distribution with cdf and pdf given by $G(x; \lambda, \gamma) = 1 - (1 - x^\lambda)^\gamma$ and $g(x; \lambda, \gamma) = \lambda\gamma x^{\lambda-1}(1 - x^\lambda)^{\gamma-1}$, respectively, for $\lambda, \gamma > 0$ and $x > 0$, as the baseline distribution, then we obtain the RB-TL-TII-EHL-Kum distribution with cdf and pdf given by

$$F_{RB-TL-TII-EHL-Kum}(x; \delta, a, b, \lambda, \gamma) = 1 - \frac{\gamma\left(\delta, -\log\left(\left[1 - \left(\frac{(1-x^\lambda)^\gamma}{1 + \left(1 - (1-x^\lambda)^\gamma\right)}\right)^{2a}\right]^b\right)\right)}{\Gamma(\delta)},$$

and

$$\begin{aligned} f_{RB-TL-TII-EHL-Kum}(x; \delta, a, b, \lambda, \gamma) &= \frac{4ab}{\Gamma(\delta)}\left[-\log\left(\left[1 - \left(\frac{(1-x^\lambda)^\gamma}{1 + \left(1 - (1-x^\lambda)^\gamma\right)}\right)^{2a}\right]^b\right)\right]^{\delta-1} \\ &\times \left[1 - \left(\frac{(1-x^\lambda)^\gamma}{1 + \left(1 - (1-x^\lambda)^\gamma\right)}\right)^{2a}\right]^{b-1}\left(\left(1-x^\lambda\right)^\gamma\right)^{2a-1} \\ &\times \frac{\lambda\gamma x^{\lambda-1}(1-x^\lambda)^{\gamma-1}}{\left[1 + \left(1 - (1-x^\lambda)^\gamma\right)\right]^{2(a+1)-1}}, \end{aligned}$$

for $\delta, a, b, \lambda, \gamma > 0$.

Figure 5 shows the plots of the pdf and the hrf of the RB-TL-TII-EHL-Kum distribution for different parameter values. The pdf can take several shapes including right-skewed, left-skewed, U-shape and reverse-J shape. The hrf of the RB-TL-TII-EHL-Kum distribution displays increasing, decreasing, bathtub, and upside-down bathtub followed by bathtub shapes.

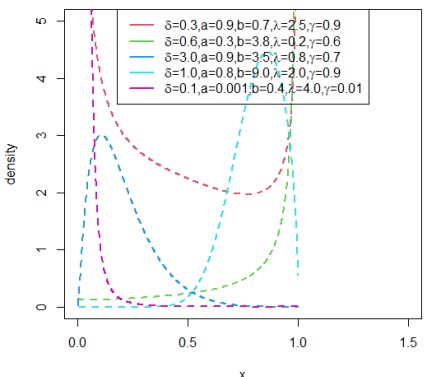 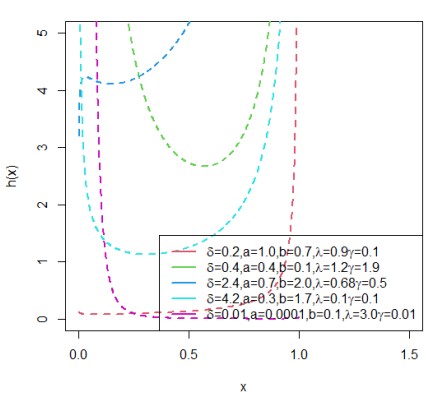

**Figure 5.** Plots of the pdf and hrf of the RB-TL-TII-EHL-Kum distribution.

## 7. Applications

In this section, three real datasets are analyzed to illustrate the flexibility, importance and modeling ability of the RB-TL-TII-EHL-G family of distibutions by using its special case, namely RB-TL-TII-EHL-W distribution. We also compare the RB-TL-TII-EHL-W distribution with the well-known models generated via the gamma transformation and modified forms of Weibull models in the literature including the gamma odd power generalized Weibull-Lomax (RB-OPGW-Lx) distribution by [23], the gamma-generalized inverse Weibull (GGIW) distribution by [24], the gamma odd Burr III-log-logistic (RB-BII-LLoG) distribution by [26], the Topp-Leone-Marshall–Olkin Weibull (TLMOW) distribution by [39], the alpha power Topp-Leone-Weibull (APTLW) distribution by [40], the type II exponentiated half logistic Weibull (TIIEHLW) distribution by [18], the odd generalized half logistic Weibull–Weibull (OGHLW-W) distribution by [41], the Marshall–Olkin exponential Weibull (MOEW) distribution by [42], and the odd Lomax generalized exponential (OLGE) distribution by [43].

To compare the fitted models, we used well-known goodness-of-fit statistics such as -2log-likelihood statistic ($-2\ln(L)$), Akaike Information Criterion ($AIC = 2p - 2\ln(L)$), Consistent Akaike Information Criterion ($CAIC = AIC + 2\frac{p(p+1)}{n-p-1}$), Bayesian Information Criterion ($BIC = p\ln(n) - 2\ln(L)$) ($n$ is the number of observations, and $p$ is the number of estimated parameters), Cramér-von Mises statistic ($W^*$), Anderson–Darling statistics ($A^*$), Kolmogorov–Smirnov (K-S) statistic, and its $p$-value. Generally, a fitted distribution can be considered as the proper fitting model for a certain dataset if it is asscociated with smaller values of all the goodness-of-fit statistics except for the $p$-value of the K-S statistic.

### 7.1. COVID-19 Data

These data are about COVID-19 collected from Holland. The data were analyzed by [44,45]. This dataset consists of 30 observations and is recorded between 31 March 2020 and 30 April 2020. (See the data in the Appendix A).

Figure 6 shows the profile likelihood plots for parameters of the RB-TL-TII-EHL-W distribution on the COVID-19 data. It can be seen that the MLEs obtained for the RB-TL-TII-EHL-W distribution are unique. This shows that the parameters of the RB-TL-TII-EHL-W distribution are identifiable.

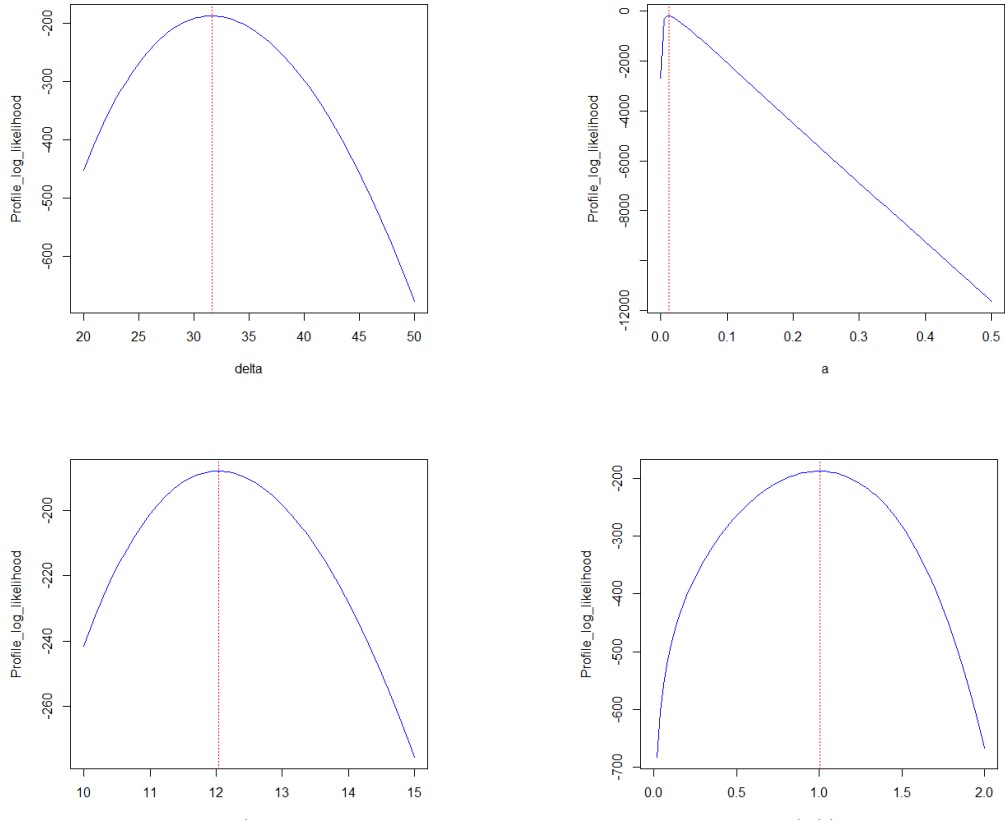

**Figure 6.** Profile likelihood function plots for parameters of RB-TL-TII-EHL-W distribution on the COVID-19 data.

The data analysis results for the COVID-19 data are given in Table 3. Table 3 indicates that the RB-TL-TII-EHL-W distribution has the lowest values of the $-2\ln(L), AIC, CAIC,$ $BIC, W^*, A^*, K - S$ and largest $p$-value of the $K - S$ statistic among other fitted models. This implies that the RB-TL-TII-EHL-W can be chosen as the best model for modeling the COVID-19 data. The fitted density plots and the probability plots (Figure 7) show that the RB-TL-TII-EHL-W distribution adequately fits the COVID-19 data.

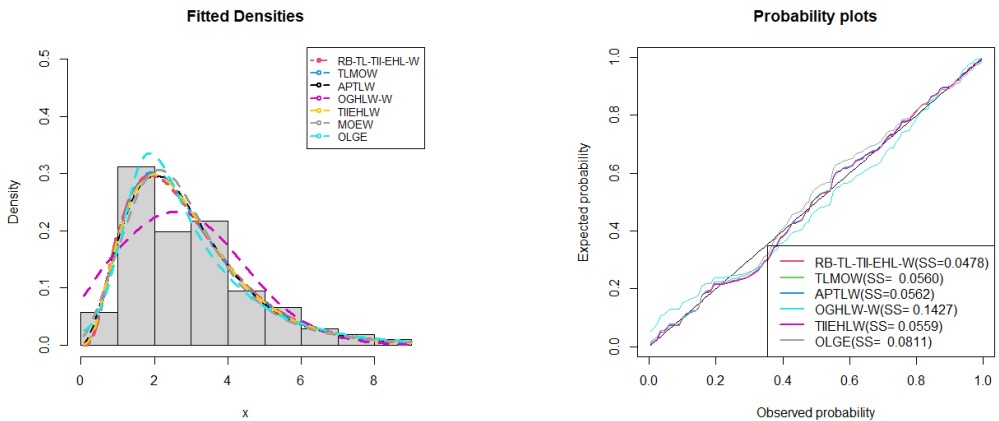

**Figure 7.** Fitted density superimposed on the histogram and observed probability vs. expected probability plots for the COVID-19 data.

**Table 3.** MLEs and Goodness-of-Fit Statistics of COVID-19 Data.

| Model | Estimates | | | | Statistics | | | | | | | |
|---|---|---|---|---|---|---|---|---|---|---|---|---|
| | $\delta$ | a | b | $\lambda$ | $-2\log L$ | AIC | AICC | BIC | W* | A* | K-S | p-Value |
| RB-TL-TII-EHL-W | 31.5910 $(7.4281 \times 10^{-04})$ | 0.0119 $(8.8791 \times 10^{-04})$ | 12.0320 $(4.3483 \times 10^{-03})$ | 1.0019 $(6.3719 \times 10^{-02})$ | 375.7258 | 383.7257 | 384.1218 | 394.3795 | 0.0460 | 0.2574 | 0.0646 | 0.7669 |
| RB-OPGW-Lx | $\alpha$ 2.3656 $(1.6820 \times 10^{-01})$ | $\beta$ 53.9060 $(1.3300 \times 10^{-04})$ | $\delta$ 1.0790 $(\times 10^{-15})$ | $k$ 0.0999 $(1.0572 \times 10^{-02})$ | 383.0791 | 391.0791 | 391.4751 | 401.7328 | 0.1156 | 0.7575 | 0.0733 | 0.6187 |
| GGIW | $k$ 7.3731 $(0.0012)$ | $\beta$ 0.2671 $(0.0259)$ | $\lambda$ 0.0026 $(0.00048)$ | $\delta$ 1.5621 $(0.0035)$ | 376.2939 | 384.2939 | 384.6900 | 394.9477 | 0.0515 | 0.2869 | 0.0670 | 0.7269 |
| RB-BIII-LLoG | $\alpha$ 0.0146 $(1.0707 \times 10^{-03})$ | $\beta$ 36.9600 $(2.8804 \times 10^{-01})$ | $\delta$ 18.3780 $(4.9356 \times 10^{-01})$ | $\lambda$ 34.8890 $(6.2419 \times 10^{-04})$ | 378.785 | 386.7852 | 387.1812 | 397.439 | 0.0759 | 0.4398 | 0.0706 | 0.6656 |
| TLMOW | $b$ 2.5861 $(2.3351)$ | $\delta$ 0.5294 $(0.6827)$ | $\lambda$ 0.1205 $(0.2658)$ | $\gamma$ 1.3825 $(0.7581)$ | 376.4293 | 384.4293 | 384.8254 | 395.0831 | 0.0535 | 0.2981 | 0.0678 | 0.7133 |
| APTLW | $\theta$ 5.8553 $(4.5983 \times 10^{-03})$ | $\alpha$ $2.7609 \times 10^{-05}$ $(2.2908 \times 10^{-04})$ | $\beta$ 0.5942 $(0.0985)$ | $\lambda$ 0.2816 $(0.0475)$ | 377.9029 | 385.903 | 386.299 | 396.5567 | 0.0637 | 0.3624 | 0.0673 | 0.7215 |
| TIIEHLW | $a$ 21.7121 $(37.3034)$ | $\lambda$ 197.2789 $(1.0424)$ | $\delta$ 3.4401 $(0.2509)$ | $\gamma$ 0.1250 $(0.0413)$ | 376.5885 | 384.5885 | 384.9845 | 395.2423 | 0.0540 | 0.3030 | 0.0693 | 0.6885 |
| OGHLW | $\alpha$ $2.4469 \times 10^{-05}$ $(4.6219 \times 10^{-06})$ | $\beta$ 1.0482 $(1.0925 \times 10^{-03})$ | $\lambda$ 8.8095 $(1.2995 \times 10^{-04})$ | $\gamma$ 0.1456 $(9.3622 \times 10^{-03})$ | 390.3392 | 398.3391 | 398.7351 | 408.9928 | 0.1887 | 1.2185 | 0.0801 | 0.5044 |
| MOEW | $\alpha$ $2.8837 \times 10^{07}$ $(9.7681 \times 10^{-09})$ | $\lambda$ $1.7441 \times 10^{-01}$ $(2.4798 \times 10^{-01})$ | $\beta$ $1.4516 \times 10^{01}$ $(2.8610 \times 10^{-01})$ | $k$ $1.4997 \times 10^{-01})$ $(3.6792 \times 10^{-02})$ | 380.8861 | 388.8859 | 389.2819 | 399.5396 | 0.0912 | 0.528 | 0.0722 | 0.6367 |
| OGLE | $\delta$ 106.3274 $(158.6800)$ | $\alpha$ 0.3341 $(0.4618)$ | $b$ 2.7092 $(0.8918)$ | $\theta$ 0.2189 $(0.0922)$ | 380.4094 | 388.4094 | 388.8054 | 399.0631 | 0.0869 | 0.4843 | 0.0738 | 0.6097 |

Figure 8 shows the observed and the fitted Kaplan–Meier (K-M) survival curves, theoretical and empirical cumulative distribution (ECDF), total test on time (TTT) scaled plots, and hazard rate function (hrf) plots. We can see that the RB-TL-TII-EHL-W distribution follows the empirical cdf and Kaplan–Meier survival curves very closely. The TTT scaled plot shows an increasing hrf. Furthermore, the estimated hrf in is agreement with the TTT scaled plot, as it also displays an increasing shape for the COVID-19 dataset.

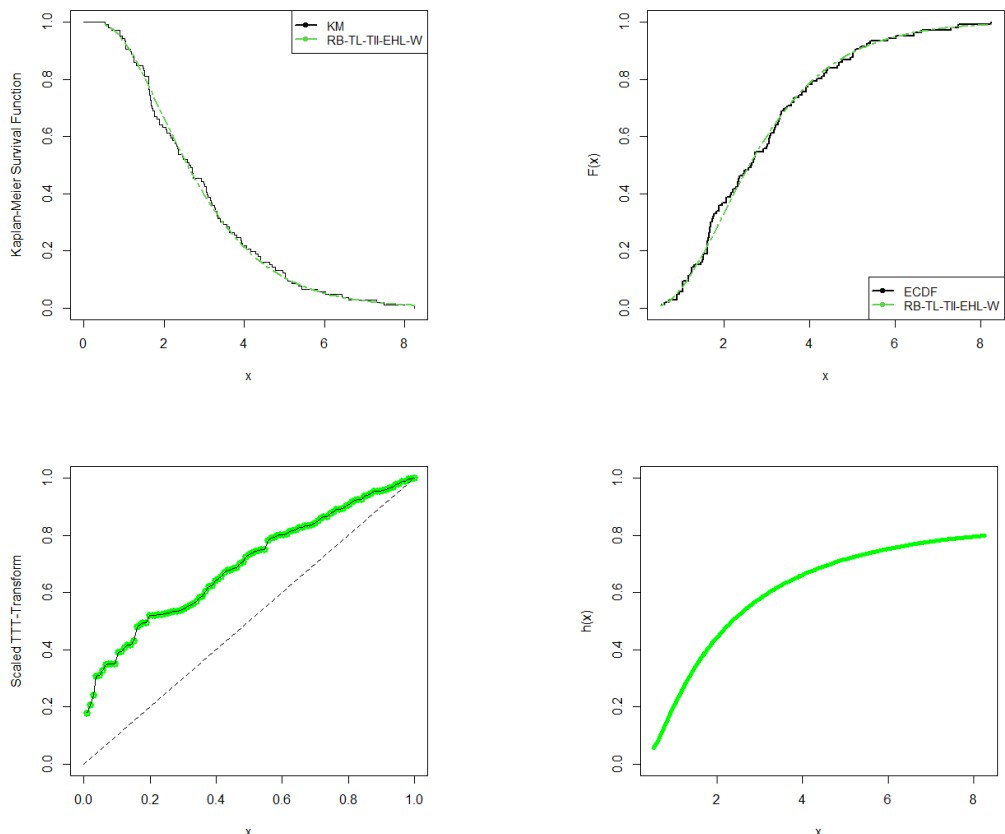

**Figure 8.** Fitted Kaplan–Meier survival curve, empirical cumulative distribution functions, the total time on the test scaled plot, and the fitted hazard rate function for the COVID-19 data.

*7.2. Vehicle Fatalities Data*

The data consist of the number of vehicle fatalities for 39 cities in South Carolina for 2012 collected by the National Highway Traffic Safety Administration (www-fars.nhtsa.dot.gov/States, accessed on 11 May 2023). The data were analyzed by [46] (see the data in Appendix A).

Figure 9 shows the profile likelihood plots for parameters of the RB-TL-TII-EHL-W distribution on the vehicle fatalities data. From the plots, we can see that the MLEs for the parameters of the RB-TL-TII-EHL-W distribution are achieved at single and different points.

Table 4 gives the MLEs of the fitted distributions together with the standard errors (in parenthesis) and all the values of the considered goodness-of-fit statistics. It is evident that the RB-TL-TII-EHL-W distribution provides the best fit among the competitors, since it has the lowest value of $-2\ln(L)$, $AIC$, $CAIC$, $BIC$, $W^*$, $A^*$ and K-S statistic. Furthermore, the $p$-value of the $K - S$ statistic is the largest for the new model, suggesting that the proposed RB-TL-TII-EHL-W model provides the best fit for the vehicle fatalities data. The fitted density plots and the probability plots (Figure 10) show that the RB-TL-TII-EHL-W distribution adequately fits the vehicle fatalities data.

**Table 4.** MLEs and Goodness-of-Fit Statistics of Vehicle Fatalities Data.

| Model | Estimates | | | | Statistics | | | | | | | |
|---|---|---|---|---|---|---|---|---|---|---|---|---|
| | $\delta$ | a | b | $\lambda$ | $-2\log L$ | AIC | AICC | BIC | W* | A* | K-S | *p*-Value |
| RB-TL-TII-EHL-W | 12.2752 (0.0018) | 0.0057 (0.0014) | 5.3660 (0.0095) | 0.8217 (0.0826) | 305.9453 | 313.9453 | 315.1218 | 320.5995 | 0.0368 | 0.2677 | 0.0886 | 0.9192 |
| RB-OPGW-Lx | $\alpha$ 3.9177 $(4.3035 \times 10^{-01})$ | $\beta$ $5.1996 \times 10^{-02}$ $(2.8295 \times 10^{-02})$ | $\delta$ $2.6947 \times 10^{01}$ $(2.5178 \times 10^{-01})$ | $k$ $1.0968 \times 10^{-03}$ $(4.6193 \times 10^{-04})$ | 380.0707 | 388.0708 | 389.2473 | 394.7251 | 0.1403 | 0.8925 | 0.3341 | 0.0003 |
| GGIW | $k$ $1.6099 \times 10^{-04}$ $(1.7939 \times 10^{-04})$ | $\beta$ 1.0006 $(5.3681 \times 10^{-04})$ | $\lambda$ $8.0186 \times 10^{-02}$ $(7.9390 \times 10^{-03})$ | $\delta$ $1.4139 \times 10^{02}$ $(2.6831 \times 10^{-06})$ | 309.3782 | 317.378 | 318.5545 | 324.0322 | 0.0606 | 0.42001 | 0.1132 | 0.6993 |
| RB-BIII-LLoG | $\alpha$ $8.1979 \times 10^{01}$ $(8.0682 \times 10^{-05})$ | $\beta$ $9.4819 \times 10^{-02}$ $(7.7779 \times 10^{-02})$ | $\delta$ $4.0700 \times 10^{-02}$ $(8.4369 \times 10^{-03})$ | $\lambda$ $8.6497 \times 10^{-02}$ $(1.4133 \times 10^{-02})$ | 709.0801 | 717.0806 | 718.257 | 723.7348 | 0.2085 | 1.3047 | 0.3004 | 0.0017 |
| TLMOW | $b$ 35.1447 (0.0010) | $\delta$ 0.2747 (0.2166) | $\lambda$ 0.3352 (0.2353) | $\gamma$ 0.4224 (0.0900) | 310.2705 | 318.2705 | 319.447 | 324.9248 | 0.0716 | 0.4849 | 0.1230 | 0.5966 |
| APTLW | $\theta$ $5.8809 \times 10^{01}$ $(5.2275 \times 10^{-04})$ | $\alpha$ $4.7573 \times 10^{-04}$ $(2.4522 \times 10^{-03})$ | $\beta$ $1.5583 \times 10^{-01}$ $(3.4123 \times 10^{-02})$ | $\lambda$ 1.0793 $(6.4652 \times 10^{-02})$ | 308.4781 | 316.4766 | 317.653 | 323.1308 | 0.0402 | 0.3019 | 0.1178 | 0.6514 |
| TIIEHLW | $a$ 0.0597 (0.0157) | $\lambda$ 0.0789 (0.0452) | $\delta$ 3.0136 (0.7527) | $\gamma$ 0.5026 (0.0691) | 360.9537 | 368.9538 | 370.1302 | 375.608 | 0.0371 | 0.2727 | 0.3485 | 0.0002 |
| OGHLW | $\alpha$ $2.3879 \times 10^{-05}$ $(9.7110 \times 10^{-06})$ | $\beta$ 0.6967 $(1.0280 \times 10^{-03})$ | $\lambda$ 11.8160 $(6.0584 \times 10^{-05})$ | $\gamma$ 0.0952 $(9.2143 \times 10^{-03})$ | 309.7039 | 317.7039 | 318.8804 | 324.3582 | 0.1143 | 0.7240 | 0.1221 | 0.6053 |
| MOEW | $\alpha$ 54.1287 (49.8948) | $\lambda$ 0.0192 (0.0476) | $\beta$ 1.3654 (0.7965) | $k$ 0.3773 (0.1854) | 309.2931 | 317.2931 | 318.4696 | 323.9473 | 0.0670 | 0.4676 | 0.0973 | 0.8539 |
| OGLE | $\delta$ 0.1170 (0.3395) | $\alpha$ 10.025 (18.7865) | $b$ 0.4439 (0.3848) | $\theta$ 0.1237 (0.1274) | 315.6349 | 323.6372 | 324.8136 | 330.2914 | 0.1351 | 0.8479 | 0.1408 | 0.4217 |

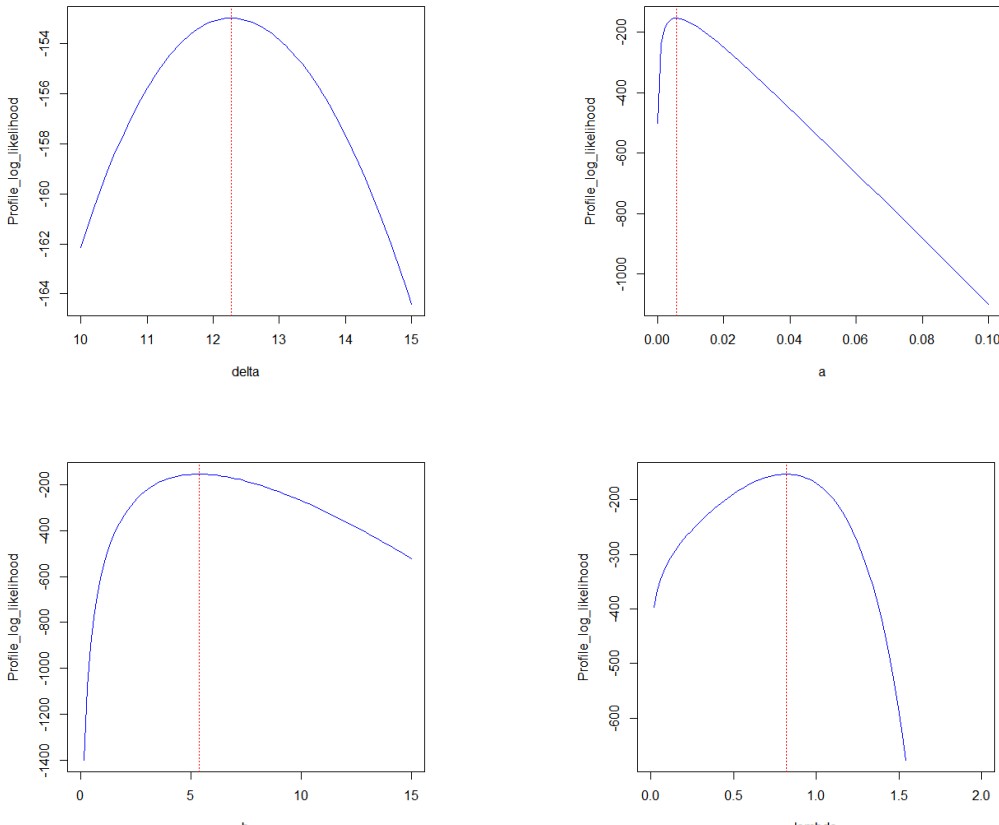

**Figure 9.** Profile likelihood function plots for parameters of RB-TL-TII-EHL-W distribution on the vehicle fatalities data.

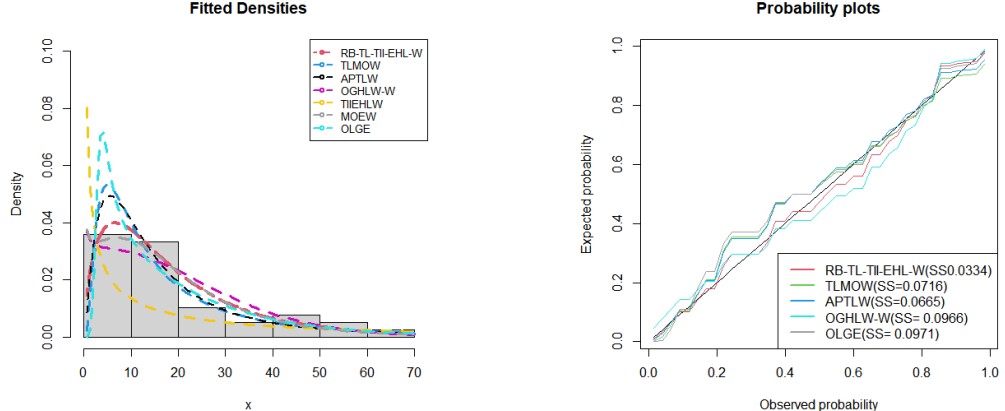

**Figure 10.** Fitted density superimposed on the histogram and observed probability vs. expected probability plots for the vehicle fatalities data.

Figure 11 shows the observed and the fitted Kaplan–Meier survival curves, ECDF plots, TTT scaled plot and hrf plot. We can see that the RB-TL-TII-EHL-W distribution follows the empirical cdf and Kaplan–Meier survival curves very closely. The TTT scaled plot and hrf plot show that the hrf for the vehicle fatalities data is non-monotonic.

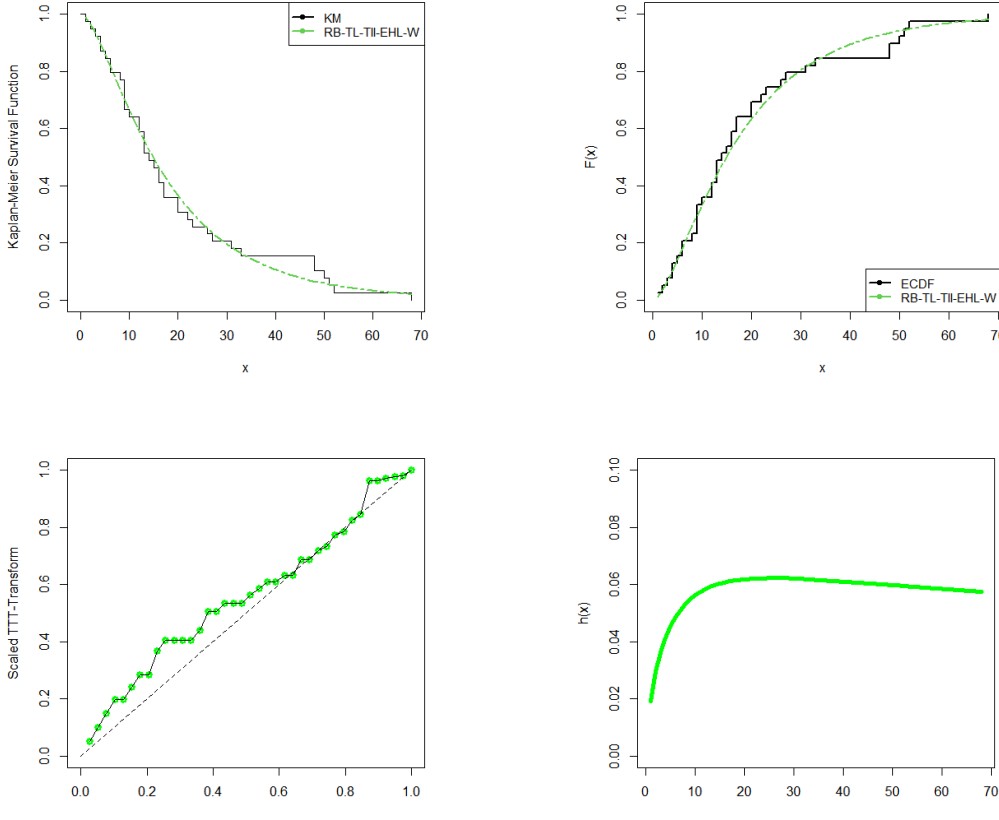

**Figure 11.** Fitted Kaplan–Meier survival curve, empirical cumulative distribution functions, the total time on the test scaled plot, and the fitted hazard rate function for the vehicle fatalities data.

*7.3. Remission Times Data*

The dataset is on remission times (months) of 128 bladder cancer patients by [47] (see the data in Appendix A).

The profile likelihood plots for parameters of the RB-TL-TII-EHL-W distribution can be used to check for the identifiability of parameters. From the plots in Figure 12, we can see that the MLEs for the RB-TL-TII-EHL-W distribution are unique; hence, we conclude that the parameters are identifiable.

The results are presented in Table 5. From Table 5, it is quite visible that the values of $-2\ln(L)$, $AIC$, $CAIC$, $BIC$, $W^*$, $A^*$, $K - S$ are smaller and the *p*-value of the K-S statistic is largest under the RB-TL-TII-EHL-W distribution, indicating that the RB-TL-TII-EHL-W distribution fits the remission times data better than other fitted distributions. The fitted density plots and the probability plots (Figure 13) show that the RB-TL-TII-EHL-W distribution adequately fits the remission times.

The observed and the fitted Kaplan–Meier survival curves, observed and fitted ECDF, TTT scaled plots and hrf plots of the RB-TL-TII-EHL-W distribution are shown in Figure 14. From the Kaplan–Meier and ECDF plots, it is clear that the RB-TL-TII-EHL-W distribution is a good candidate for modeling the remission times data. The TTT scaled and hrf plots indicate that indeed, the RB-TL-TII-EHL-W distribution is suitable for modeling the remission times data, as they both estimate the hazard rate of the data to be upside-down bathtub.

**Table 5.** MLEs and Goodness-of-Fit Statistics of Remission Times Data.

| Model | Estimates | | | | Statistics | | | | | | | |
|---|---|---|---|---|---|---|---|---|---|---|---|---|
| | $\delta$ | a | b | $\lambda$ | $-2\log L$ | AIC | AICC | BIC | W* | A* | K-S | *p*-Value |
| RB-TL-TII-EHL-W | 24.6260 (0.0015) | 0.00010 ($1.5862 \times 10^{-05}$)) | 4.0077 (0.0189) | 1.2704 (0.0665) | 822.8816 | 830.8816 | 831.2068 | 842.2897 | 0.0579 | 0.3833 | 0.0522 | 0.8752 |
| | $\alpha$ | $\beta$ | $\delta$ | k | | | | | | | | |
| RB-OPGW-Lx | 1.8034 ($1.2081 \times 10^{-01}$) | 592.3000 ($2.1182 \times 10^{-06}$) | 1.0001 ($1.8869 \times 10^{-15}$) | 0.0100 ($2.2921 \times 10^{-03}$) | 825.8694 | 833.8694 | 834.1946 | 845.2776 | 0.0916 | 0.5778 | 0.0690 | 0.5744 |
| | k | $\beta$ | $\lambda$ | $\delta$ | | | | | | | | |
| GGIW | 12.7271 (1.0673) | 0.1990 (0.0206) | 0.0016 (0.0012) | 0.2197 (0.0241) | 877.5066 | 836.542 | 836.8672 | 847.9502 | 0.1316 | 0.8126 | 0.0702 | 0.552 |
| | $\alpha$ | $\beta$ | $\delta$ | $\lambda$ | | | | | | | | |
| RB-BIII-LLoG | 0.2066 (0.2573) | 35.6475 (23.6792) | 17.4118 (13.7133) | 1.3549 (1.6875) | 832.1743 | 840.1743 | 840.4995 | 851.5824 | 0.1428 | 0.9589 | 0.0676 | 0.6004 |
| | b | $\delta$ | $\lambda$ | $\gamma$ | | | | | | | | |
| TLMOW | 16.1350 (32.4118) | 0.2540 (0.1033) | 0.3205 (0.4540) | 0.4398 (0.2468) | 837.4597 | 845.4595 | 845.7847 | 856.8676 | 0.1979 | 1.2993 | 0.0811 | 0.3679 |
| | $\theta$ | $\alpha$ | $\beta$ | $\lambda$ | | | | | | | | |
| APTLW | 0.3632 (0.1239) | $2.6272 \times 10^{02}$ ($4.8767 \times 10^{-05}$) | 0.8610 (0.1241) | 0.1120 (0.0540) | 827.4507 | 835.4507 | 835.7759 | 846.8588 | 0.1275 | 0.7620 | 0.0727 | 0.5065 |
| | a | $\lambda$ | $\delta$ | $\gamma$ | | | | | | | | |
| TIIEHLW | $1.0297 \times 10^{03}$ ($1.8761 \times 10^{-06}$) | $1.0759 \times 10^{02}$ ($1.9066 \times 10^{-04}$) | 2.4004 ($2.1528 \times 10^{-02}$) | 0.0500 ($3.2111 \times 10^{-03}$) | 823.0934 | 831.0934 | 831.4186 | 842.5015 | 0.0710 | 0.4392 | 0.0549 | 0.8351 |
| | $\alpha$ | $\beta$ | $\lambda$ | $\gamma$ | | | | | | | | |
| OGHLWW | $2.1269 \times 10^{-05}$ ($3.4793 \times 10^{-06}$) | 0.6471 ($4.2738 \times 10^{-04}$) | 14.4550 ($1.9132 \times 10^{-05}$) | 0.0774 ($4.3136 \times 10^{-03}$) | 838.0349 | 846.0347 | 846.3599 | 857.4428 | 0.2477 | 1.4603 | 0.0952 | 0.1962 |
| | $\alpha$ | $\lambda$ | $\beta$ | k | | | | | | | | |
| MOEW | 2.3544 (1.2446) | 0.0714 (0.0372) | 1.0415 (1.0080) | 0.2405 (0.2691) | 948.2204 | 956.2204 | 956.5456 | 967.6285 | 0.2321 | 1.3721 | 0.4481 | 0.0000 |
| | $\delta$ | $\alpha$ | b | $\theta$ | | | | | | | | |
| OGLE | 8.8103 (8.3076) | 0.2220 (0.1826) | 0.3397 (0.0966) | 0.3668 (0.1147) | 836.6097 | 844.6097 | 844.9349 | 856.0178 | 0.2298 | 1.3899 | 0.0880 | 0.2745 |

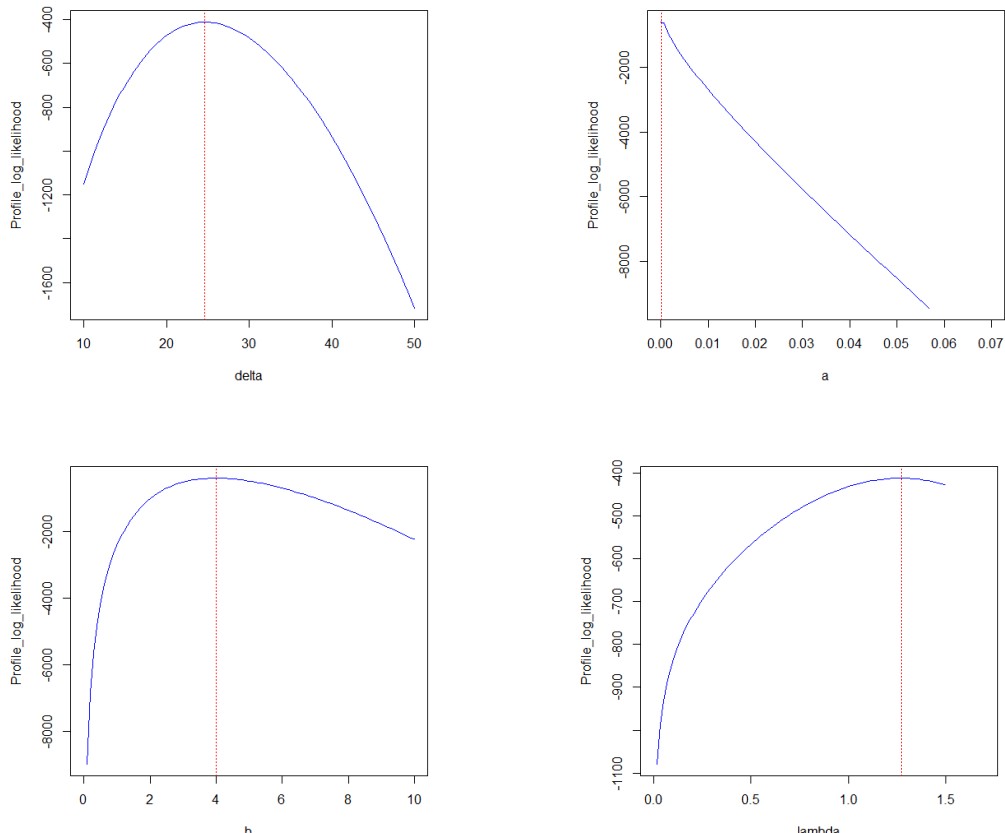

**Figure 12.** Profile likelihood function plots for parameters of RB-TL-TII-EHL-W distribution on the remission times dataset.

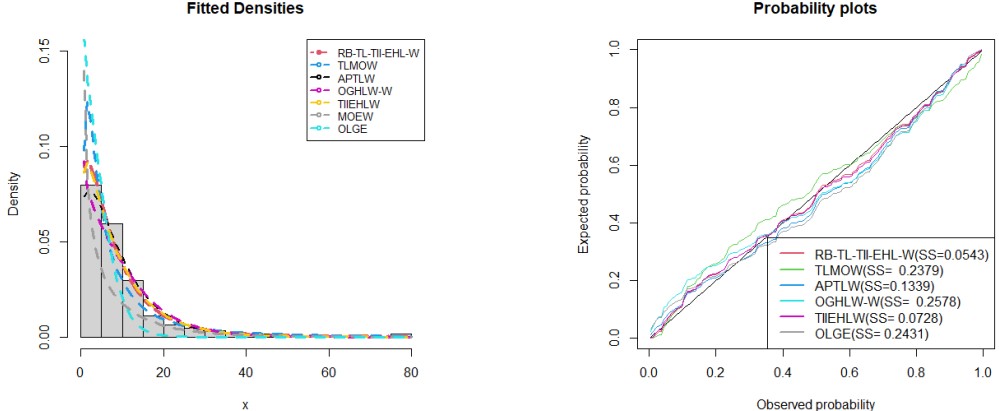

**Figure 13.** Fitted density superimposed on the histogram and observed probability vs. expected probability plots for the remission times data.

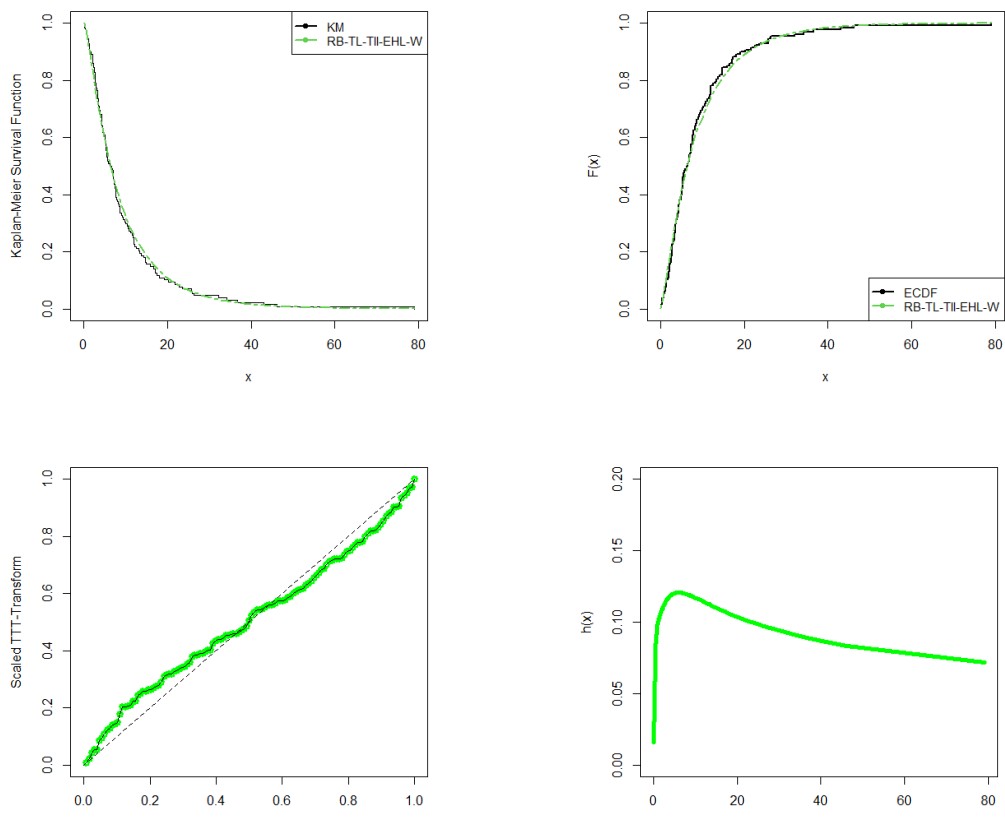

**Figure 14.** Fitted Kaplan–Meier survival curve, empirical cumulative distribution functions, the total time on the test scaled plot, and the fitted hazard rate function for the remission times data.

## 8. Conclusions

We have proposed and developed a new generalized family of distributions called the Gamma-Topp-Leone-Type II-Exponentiated Half Logistic-G (RB-TL-TII-EHL-G) distribution. It has been shown that this new family of distributions can be expressed as an infinite linear combination of the exponentiated-G distributions. The maximum likelihood estimation technique was used to estimate the model parameters. Some of the mathematical and statistical properties have been derived and established. The RB-TL-TII-EHL-W model as a special case to this new family of distributions was applied to three datasets, and from the results, it is evident that the new proposed model performs better than several equal-parameter models. In the future, we will seek to use different estimation methods to estimate the unknown parameters and also also apply the new distribution to censored data.

**Author Contributions:** Conceptualization, B.O.; Methodology, B.O.; Software, T.M.; Formal analysis, B.O.; Writing—review & editing, B.O. and T.M. All authors have read and agreed to the published version of the manuscript.

**Funding:** This research received no external funding.

**Institutional Review Board Statement:** Not Applicable.

**Informed Consent Statement:** Not Applicable.

**Data Availability Statement:** Not Applicable.

**Conflicts of Interest:** The authors declare no conflict of interest.

## Appendix A

Click on the link below for results in the appendix. https://drive.google.com/file/d/1Hpx6TJPNKBAcyP71BTaHjWGBJ6orInmI/view?usp=share_link, accessed on 11 May 2023.

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
