# Peer review of "The Gamma-Topp-Leone-Type II-Exponentiated Half Logistic-G Family of Distributions with Applications"

_stats, doi:10.3390/stats6020045_

Round 1

Reviewer 1 Report

This work derives and studies some of the main statistical characteristics of (RB-TL-TII-EHL-G) family of distributions.  In my opinion, this paper is written too long and it is better to be more concise 

Author Response

Please see attached author's response to reviewer 1.

Reviewer 2 Report

English language should be revised well

Author Response

Please see attached author's response to reviewer 2

Reviewer 3 Report

Comments to “The Gamma-Topp-Leone-Type II-Exponentiated Half Logistic-G Family of Distributions with Applications

Overall, the quality of the presentation must be improved, need some explanation:

1.      In Abstract should present the general conclusion of the corresponding research.

2.      Explain (or cite(s) the corresponding literatures) why did you define Eq. (5) and (6), (12), and (13).

3.      Delete 3.1, since there is no 3.2.

4.      In Table 1, first line, what are (0.4, 2.0, 1.0, 0.6) and (1.0, 1.0, 2.0, 0.6) mean?

5.      Why did you use maximum likelihood estimation to estimate the parameters? What do guarantees all classical assumptions fulfilled?

6.      What is Renyi Entropi?

 Minor editing of English language required

Author Response

Please see attached author's response to reviewer 3

Reviewer 4 Report

The authors introduced the new Risti´c and Balakhrisnan or Gamma-Topp-Leone-Type II-Exponentiated Half-Logistic-G family of distributions, discussed some properties and application. I list my comments as follows,

1. The motivation you mentioned was that the flexibility of the new family of distributions to model both monotonic and non-monotonic hazard rate functions by capturing different shapes. However, these reasons could be This motivation for any new models. You need to specify the significance of this new distribution, otherwise, there would be another new distribution based on your proposed one. 

2. There are some structure issues, such as, Line 57 needs some spaces; Line 100 [34]?. Moreover, page 5, put these calculations into appendix or simplify them, the same comments for page 9. 

3. In the subsection, you mentioned the detailed steps could be put into the appendix, however, I did not find the appendix in this manuscript.

4. In your simulation part, the accuracy of some parameters is not well, i.e., for λ and b, even for the large sample size. How to fixed this issue? 

5. You give some special distributions from this family; how do the people know which special distribution to choose when solving a practical problem?

Author Response

Please see attached response to reviewer's report

Round 2

Reviewer 4 Report

Thank you.